# Anti-CD19 CAR T cells potently redirected to kill solid tumor cells

Christine Ambrose[1], Lihe Su[1], Lan Wu[1], Fay J. Dufort[1], Thomas Sanford[1], Alyssa Birt[1], Benjamin J. Hackel[2], Andreas Hombach[3], Hinrich Abken[3¤], Roy R. Lobb[1], Paul D. Rennert[1]*

1 Aleta Biotherapeutics, Natick, MA, United States of America, 2 University of Minnesota, Minneapolis, MN, United States of America, 3 University of Cologne, Köln, Germany

¤ Current address: Regensburg University, Regensburg, Germany
* paul.rennert@aletabio.com

**Data Availability Statement:** All relevant data are within the manuscript and its Supporting Information files.

**Funding:** Aleta Biotherapeutics is wholly funded by Advent Life Science Partners, a venture capital

## Abstract

Successful CAR T cell therapy for the treatment of solid tumors requires exemplary CAR T cell expansion, persistence and fitness, and the ability to target tumor antigens safely. Here we address this constellation of critical attributes for successful cellular therapy by using integrated technologies that simplify development and derisk clinical translation. We have developed a CAR-CD19 T cell that secretes a CD19-anti-Her2 bridging protein. This cell therapy strategy exploits the ability of CD19-targeting CAR T cells to interact with CD19 on normal B cells to drive expansion, persistence and fitness. The secreted bridging protein potently binds to Her2-positive tumor cells, mediating CAR-CD19 T cell cytotoxicity *in vitro* and *in vivo*. Because of its short half-life, the secreted bridging protein will selectively accumulate at the site of highest antigen expression, ie. at the tumor. Bridging proteins that bind to multiple different tumor antigens have been created. Therefore, antigen-bridging CAR-CD19 T cells incorporate critical attributes for successful solid tumor cell therapy. This platform can be exploited to attack tumor antigens on any cancer.

## Introduction

The treatment of relapsed or refractory Acute Lymphocytic Leukemia (ALL) and Non-Hodgkin Lymphoma (NHL) with chimeric antigen receptor (CAR) T-cells that target CD19 (CAR-CD19 T cells) has led to FDA and EMA approvals of the adoptive cell therapies tisagenlecleucel and axicabtagene ciloleucel [1, 2]. CAR-CD19 T cells have thus paved the way for the development of cellular therapeutics as 'living drugs', and have driven our understanding of CAR T cell regulatory, CMC and commercialization issues [3]. Furthermore, CAR-CD19 T cells are routinely used to evaluate all aspects of CAR T cell function, including optimization of signaling domains and hinge regions [4], development of dual CARs [5, 6], expression of cytokines, antibodies, and other mediators [7, 8], evaluation of toxicities [9, 10], and of switch technologies [11, 12]. CAR-CD19 T cells also provide a simple and universal solution to the issue of CAR T cell persistence. Robust *in vivo* expansion followed by prolonged CAR-T cell persistence is critical for their efficacy in the treatment of hematologic malignancies [13, 14].

firm. Therefore we would like to state the following – Advent Life Sciences did not play a role in the study design, data collection and analysis, decision to publish, or preparation of this manuscript and only provided financial support in the form of authors' salaries and/or research materials. The funder provided support in the form of salaries for authors [CA, LS, LW, FJD, AB, PDR], and paid consulting fees [RRL] but did not have any additional role in the study design, data collection and analysis, decision to publish, or preparation of the manuscript. The specific roles of these authors are articulated in the 'author contributions' section.

**Competing interests:** The authors have read the journal's policy, and the authors of the study have the following competing interests to declare: Aleta Biotherapeutics is wholly funded by Advent Life Science Partners, a venture capital firm. Advent Life Sciences provided support in the form of salaries for authors [CA, LS, LW, FJD, AB, PDR], and paid consulting fees [RRL]. Aleta Biotherapeutics funded BJH's research at the University of Minnesota for a period of time, resulting in a shared patent filing. The agreement with Univ Minnesota was for a one-time payment from Aleta to secure all of the patent rights (assigned by Dr Hackel to UMN). The patent is "CD19 VARIANTS" US Appln. No. 62/599,211; Filed: December 15, 2017. The research sponsorship has since ended and that financial relationship in no manner has influenced the work contained in this manuscript. Aleta Biotherapeutics paid consultancy fees to HA in the past. The consultancy has since ended and that financial relationship in no manner has influenced the work contained in this manuscript. This does not alter our adherence to PLOS ONE policies on sharing data and materials. There are no other products in development or marketed products associated with this research to declare.

Since CD19-positive normal B cells are constantly produced by the bone marrow in response to B cell depletion, CAR-CD19 T cells uniquely access a non-tumor dependent and self-renewing source of activating antigen. This is true even in patients who have undergone lymphodepleting chemotherapy as the bone marrow can recover and start producing B cells in as little as 28 days [15, 16]. In contrast, other CAR T cells, particularly those targeting solid tumor antigens, have largely shown poor persistence to date [17]. In short, CAR-CD19 T cells have properties which make them uniquely suited to CAR T cell therapy, come with a broad and expanding set of pre-clinical knowledge and possess an unmatched development history.

To capitalize on this deep knowledge base, we have chosen to use CAR-CD19 T cells as a unique platform solution that will allow us to target and kill any tumor. We recently described the development of bridging proteins which contain the extracellular domain (ECD) of CD19 linked to an antigen binding domain, eg. an antibody fragment, that binds to a tumor-expressed antigen [18]. The wild-type CD19 ECD was difficult to express, therefore we developed novel CD19 ECD mutants that can be linked N- or C-terminally to any protein, generating modular CD19 bridging proteins with enhanced secretion and a predicted lack of immunogenicity [18]. These rationally designed CD19 bridging proteins are capable of binding to any tumor antigen, thereby coating CD19-negative tumor cells with CD19. CD19-coated tumor cells representing diverse indications were thereby made susceptible to potent CAR-CD19 T cell-mediated cytotoxicity [18]. Of note, the CAR-CD19 domain on the T cell remains the same regardless of the antigens targeted by the bridging protein, and this minimizes complexity in CAR T cell manufacturing across our diverse programs. This technology therefore has the potential to broaden the reach of CAR-CD19 T cells beyond B cell malignancies to any tumor without the need to build many different CAR and combination CAR constructs. Given the emerging knowledge regarding the need for multi-antigen targeting in order to ensure durable responses to CAR-T therapy the need for such a simple, pragmatic and modular technology is evident.

Here we describe the use of CAR-CD19 T cells that secrete CD19 bridging proteins as a highly effective therapeutic approach to targeting hematologic cancers and solid tumors. CD19 bridging proteins are simple and modular solutions to multi-antigen targeting, antigen selectivity, and optimization of potency and safety. This technology retains the inherent advantages of CAR-CD19 T cells and leverages the development and manufacturing knowledge accumulated to date.

## Results

### CD19-anti-Her2 bridging proteins bind with high affinity to anti-CD19 and to Her2

We have previously shown that wildtype and stabilized forms of the CD19 ECD can be fused to a variety of scFvs, VHHs, and scaffold proteins such as Fibronectin Type III (Fn3) domains, with the retention of binding function, CAR-CD19 recognition and cytotoxicity [18, 19]. These CD19-containing proteins bind to both CAR-CD19 T cells and to the targeted tumor antigens, and are referred to as bridging proteins. Here we show that bridging proteins can be secreted by CAR-CD19 T cells transduced with a lentiviral vector construct that encodes both a cell surface CAR-CD19 sequence and a bridging protein sequence. Functionally, bridging proteins act as CAR T engagers, activating the CAR T cell and killing the bound tumor cell (Fig 1).

A bridging protein containing the wildtype CD19 ECD sequence and the anti-Her2 scFv was described previously [18]. Novel, stabilized forms of the CD19 ECD were also described previously [18] and one of these (CD19$_{C6.2}$) was used to create a CD19-anti-Her2 bridging

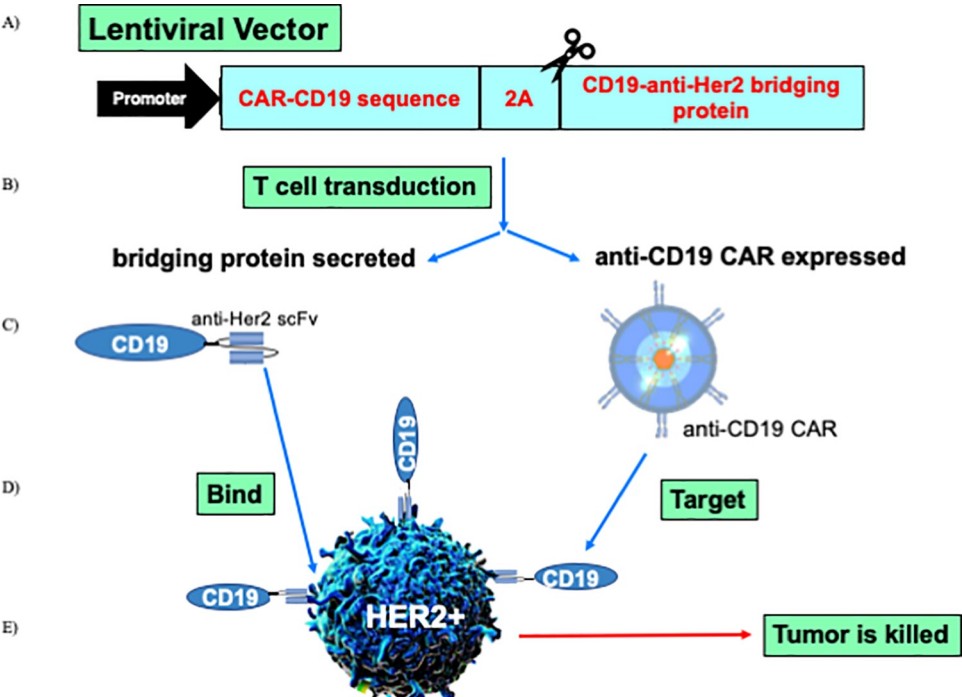

**Fig 1. Schematic representation of the technology.** A) A lentiviral expression vector was created that encodes an anti-CD19 CAR domain, a 2A cleavage site, and a secreted CD19-anti-Her2 scFv bridging protein. B) Human primary T cells were transduced and expanded. C) The T cell transduction resulted in the expression of the anti-CD19 CAR domain on the T cell surface (thus, CAR-CD19) and secretion of the bridging protein. D) The bridging protein has two domains: the anti-Her2 scFv bound to Her2 expressed on the tumor cell; the CD19 ECD bound to the anti-CD19 CAR domain on the transduced T cell. Thus the bridging protein acts as a CAR-T cell engager protein. E) The binding events created a cytotoxic immune synapse between the CAR-CD19 T cell and the Her2-positive tumor cell via recognition of CD19; this resulted in CAR T cell proliferation and tumor cell death.

protein (construct #340, Table 1). The most important difference in the two forms of the CD19 ECD used is the level of protein expression from transfected cells that these support; the stable form also binds more tightly to the anti-CD19 scFv from the FMC63 monoclonal antibody [18]. Several proteins were created as specificity controls, containing just the CD19 ECD or containing a CD22 ECD linked to the anti-Her2 scFv (constructs #28 and #117, S1 Table).

We characterized the binding properties of CD19-anti-Her2 bridging proteins. In ELISA assays CD19-bridging proteins were shown to bind both to the anti-CD19 antibody FMC63

**Table 1. List of major constructs used in the text.**

| Construct # | Description | Modality |
|---|---|---|
| 254 | CAR sequence targeting CD19 | Anti-CD19 CAR |
| 340 | Stabilized CD19 ECD–anti-Her2 | Purified BP |
| 374 | Sequence of 254 –P2A –sequence of 340 | Anti-CD19 CAR + stabilized BP |
| 390 | CAR sequence targeting Her2 | Anti-Her2 CAR |
| 460 | Anti-Her2 –stabilized CD19 ECD–anti-EGFR | Purified BP |

Constructs are presented in the numerical order with a description of the encoded sequences in column 2 and the format in which the expressed sequence is utilized in column 3. The long dashes indicate a linker sequence. P2A refers to a cleavage sequence. BP refers to a bridging protein.

**Table 2. Summary of bridging protein binding and cytotoxicity data.**

| Bridging protein binding affinity in ELISA assays ($EC_{50}$) | | | | | |
|---|---|---|---|---|---|
| **CAPTURE** | **DETECTION** | **#42** | **#340** | **#28** | **#117** |
| FMC63 | Her2 | 0.2 +/- 0.18 nM | 0.06 +/- 0.03 nM | none | none |
| | | n = 2 | n = 2 | n = 2 | n = 2 |
| Her2-Fc | FMC63 | 0.35 +/- 0.26 | 0.19 +/- 0.04 | none | none |
| | | n = 2 | n = 2 | n = 2 | n = 1 |
| Bridging protein binding affinity in flow cytometric assays ($EC_{50}$) | | | | | |
| **Cells** | CAR-CD19 T cells | 0.6 +/- 0.12 nM | 0.21 +/- 0.01 nM | - | - |
| | | n = 2 | n = 2 | | |
| | untransduced T cells | none | none | - | - |
| | | n = 2 | n = 2 | | |
| | SKOV3 | 1.2 +/- 0.74 nM | 1.5 +/- 1.1 nM | none | 0.68 +/- 0.32 nM |
| | | n = 7 | n = 2 | n = 3 | n = 2 |
| | BT474 | 0.26 +/- 0.3 nM | 0.24 +/- 0.3 nM | none | 0.45 nM |
| | | n = 3 | n = 3 | n = 2 | n = 1 |
| Bridging protein mediated cytotoxicity ($IC_{50}$) | | | | | |
| **Donor** | **Target cells** | **#42** | **#340** | **#28** | **#117** |
| 54/69 | SKOV3 | 4.8 +/- 2.7 pM | 4.7 +/- 2.6 pM | none | none |
| | | n = 14 | n = 2 | n = 2 | n = 1 |
| 54/69 | BT474 | 12.2 +/- 3.6 pM | 7.4 +/- 0.3 pM | none | - |
| | | n = 2 | n = 2 | n = 2 | |

Data are reported as mean +/- standard error. ELISA assay data were produced from triplicate wells, flow cytometry assay data were produced in duplicate, and cytotoxicity assay data were produced in triplicate. The number of repeats ('n') per experiment is shown for each assay. Top: $EC_{50}$ binding data for bridging proteins and control proteins as determined in ELISA assays. The capture reagents, detection reagents and protein construct numbers are indicated. Middle: $EC_{50}$ binding data for bridging proteins and control proteins as determined in flow cytometric assays. The cells used are indicated. Flow cytometry reagents used to detect cell-bound proteins were anti-CD19 antibody FMC63 (#42, #340, #28) or anti-His tag antibody (#117). Bottom: bridging protein-mediated CAR-CD19 cytotoxicity against Her2-positive cell lines. CAR-CD19 T cells from donor 54 (47% CAR-positive) and donor 69 (54% CAR positive) were used at an E:T ratio of 10:1. $IC_{50}$ values were derived from a titration of the bridging protein concentration in the assays. For all assays shown, 'none' indicates that no binding or activity above background was detected; a '-' indicates that the experiment was not performed.

and to the Her2 antigen with low nM affinity (Table 2, S1A and S1B Fig). We then examined binding to T cells expressing the anti-CD19 CAR domain. Donor human T cells were transduced with lentiviral particles that express the scFv derived from the anti-CD19 antibody FMC63 (CAR-CD19, construct #254, Table 1). Transduced T cells expressed the anti-CD19 CAR domain as detected with anti-Flag antibody staining (S2A Fig). We performed a flow cytometry assay to assess the binding of the CD19-anti-Her2 bridging proteins (#340, Table 1; #42, S1 Table) to anti-CD19 CAR T cells. CD19-anti-Her2 bridging proteins bound to CAR-CD19 T cells with a low nM affinity but did not bind to control (un-transduced, UTD) T cells (Table 2, S2B Fig). In the linear range of the assay (approximately 0.5–100 ng/ml) the bridging protein having a stabilized CD19 ECD (#340) demonstrated statistically superior binding (S2B Fig), consistent with previous results [18].

SKOV3 ovarian carcinoma cells and BT474 breast carcinoma cells express Her2 (S2C Fig). Flow cytometry assays using Her2-positive SKOV3 and BT474 cells demonstrated that the purified CD19-anti-Her2 bridging proteins consistently bound with low nM $EC_{50}$ values, ranging from 0.2–2.7 nM (Table 2; S2D Fig). In this binding format the bridging proteins were indistinguishable from each other, as expected given that they use the same anti-Her2 scFv from trastuzumab. These data showed that both halves of the CD19-anti-Her2 bridging protein were fully functional, binding to the CAR expressing the anti-CD19 FMC63 scFv and to

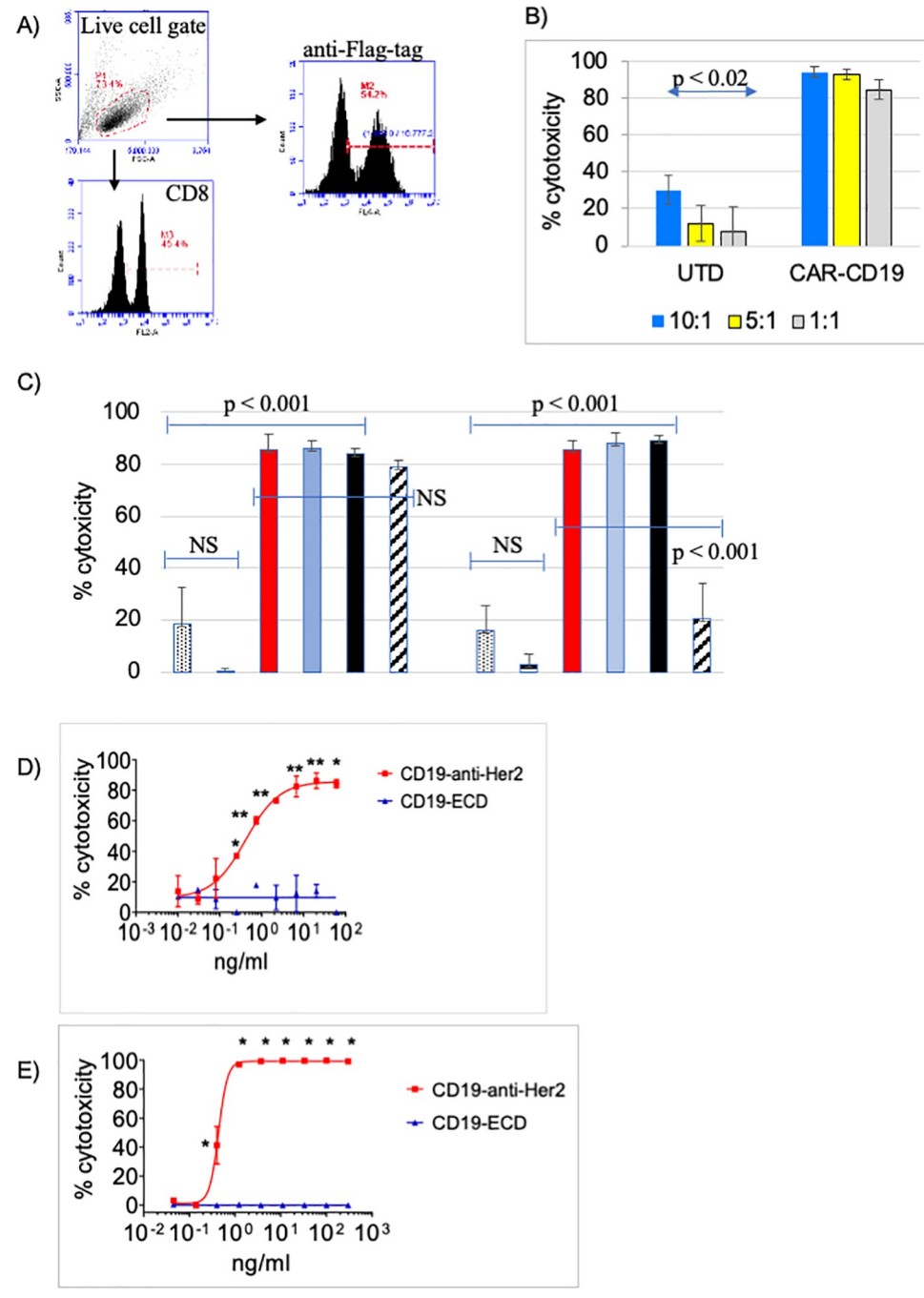

**Fig 2. A CD19-anti-Her2 bridging protein mediates potent CAR-CD19 T cell cytotoxic activity *in vitro*.** A)
Phenotype of transduced and expanded CAR-CD19 T cells (donor 45): CAR-CD19 T cells were 54% CAR-positive by
anti-Flag-tag staining. B) The CAR-CD19 T cells were tested for cytotoxicity against Nalm6 cells at different E:T ratios
(10:1, blue; 3:1, teal; 1:1 light blue) and compared to donor-matched untransduced cells (UTD) at the same E:T ratios;
all comparisons to UTD controls were significant. This assay is run routinely on all CAR-CD19-based CAR T cells; the
extent of cytotoxicity varies among donors. C) Bridging protein, CAR-CD19 T cells (donor 54, 58% Flag-positive) and
target tumor cells were added simultaneously (red bars) or bridging proteins and tumor cells were preincubated then
added to CAR-T cells (light blue bars) or bridging proteins and CAR-T cells were preincubated then added to tumor
cells (black bars). The E:T ratio used was 5:1. Controls were CAR-CD19 T cells, target cells plus bridging protein only
('BP') or target cells alone in culture ('none'). Nalm6 (left) and SKOV3 (right) cytotoxicity profiles are shown. The data
are from duplicate wells; the experiment was performed twice with similar results. D, E) Dose response cytotoxicity
curves using an E:T ratio of 10:1 CAR-CD19 T cells and target cells and varying the concentration of the CD19-anti-
Her2 bridging protein or the control CD19-ECD. Target cell lines SKOV3 (D) and BT474 (E) are shown. The

CAR-CD19 T cells used for the SKOV3 assay were from donor 54 (47% Flag-positive); donor 69 CAR-CD19 T cells (55% Flag-positive) were used for the BT474 assay. The data are from triplicate wells; the experiment was performed twice with similar results. All statistical tests were performed as 2-sided T tests. For Panels D and E, * indicates significance of $p < 0.0001$ and ** indicates significance of $p < 0.02$.

the Her2 antigen on the SKOV3 cell surface. The control CD22-anti-Her2 bridging protein bound to Her2-positive SKOV3 cells, but the CD19 ECD protein failed to bind to either SKOV3 cells or BT474 cells (Table 2, S3 Fig). These data further illustrate the specificity of the binding components within bridging proteins.

Next we turned to experiments utilizing the bridging proteins and anti-CD19 CAR T cells. In all experiments the effector to target ratio is calculated by the total number of T cells, not the number of CAR-positive T cells. Therefore the donor number and positive CAR percentage for each experiment is stated.

## CD19-anti-Her2 does not compromise CAR-CD19 function

To address the potency of direct cytotoxicity via CD19 binding, CAR-CD19 T cell activity was determined at different effector:target (E:T) ratios using the CD19-positive B cell line Nalm6. The CAR construct used is a generation 3 design (with 4-1BB/CD28/CD3 cytoplasmic components) and uses the FMC63 scFv (see Methods). The flow cytometry methods used to characterize anti-CD19 CAR T cells after transduction and expansion are shown (Fig 2A, donor 45). Cytotoxicity was routinely observed down to the lowest E:T ratio tested, although there was some variation in the extent of cytotoxicity depending on the specific normal human donor T cells used in the assay (Fig 2B).

Aberrant signaling can drive CAR T cells into a dysfunctional state [20]. Therefore we assessed the impact of binding the CD19-anti-Her2 bridging protein (#340) on CAR-CD19 T cell cytotoxicity against the target cell lines Nalm6 (CD19-positive) and SKOV3 (Her2-positive). Cytotoxicity against Nalm6 cells requires that the anti-CD19 CAR domain on the CAR T cell is able to directly kill the cells. Cytotoxicity against Her2-positive SKOV3 cells will occur when the CD19-anti-Her2 bridging protein binds to both the anti-CD19 CAR domain on the CAR T cells and to the target tumor cells, thereby forming a cytotoxic synapse. The components of the cytotoxicity assay were added in three ways: a) CAR-CD19 T cells, bridging protein and target cells were added together at same time, b) CAR-CD19 T cells were added to a premixture of bridging protein and target tumor cells, which had been pre-incubated at 37˚C for 10 minutes, or c) the CAR-CD19 T cells were premixed with bridging protein and pre-incubated at 37˚C for 10 minutes, and then added to target tumor cells. In all instances the bridging protein was added at 1 μg/ml, a concentration shown to be saturating for binding to CAR-CD19 T cells (see S2B Fig). There was no difference in the degree of cytotoxicity observed among the three different conditions on either Nalm6 cells or SKOV3 cells (Fig 2C). These experiments demonstrated that pre-binding of a CD19-anti-Her2 bridging protein does not negatively impact the ability of the CAR-CD19 T cells to directly or indirectly kill the target cells.

Next, CAR-CD19 T cell cytotoxicity curves were derived for killing Her-2 expressing SKOV3 and BT474 cells in the presence of the CD19-anti-Her2 bridging protein. The bridging proteins mediated very potent cytotoxicity with low pM $IC_{50}$ values (Table 2, Fig 2D and 2E).

## A dual-antigen-directed bridging protein potently targets Her2 and EGFR

Having successfully engineered the CD19-anti-Her2 bridging protein we added a second antigen-targeting domain that binds EGFR. The resulting bridging protein consists of anti-Her2—

**Table 3. Activity of a dual antigen targeting bridging protein.**

| Bridging protein binding affinity in flow cytometry assays (EC$_{50}$) | | | | |
|---|---|---|---|---|
| cell line | antigens expressed | anti-Her2-CD19-anti-EGFR (#460) | CD19-anti-Her2 (#311) | CD19-anti-EGFR (#416) |
| BT474 | Her2 only | 2.5 nM | 3.8 +/- 1.1 nM | - |
| | | n = 1 | n = 2 | |
| K562-EGFR | EGFR only | 0.6 nM | 0 | 0.52 nM |
| | | n = 1 | n = 2 | n = 1 |
| SKOV3 | Her2/EGFR | 0.44 nM | 2.5 +/- 0.6 nM | 0.37 nM |
| | | n = 1 | n = 3 | n = 1 |
| Bridging protein mediated cytotoxicity by CAR-CD19 T cells (IC$_{50}$) | | | | |
| cell line | antigens expressed | anti-Her2-CD19-anti-EGFR (#460) | CD19-anti-Her2 (#311) | CD19-anti-EGFR (#416) |
| BT474 | Her2 only | 5.1 pM | 24.4 +/- 3.7 pM | 0 |
| | | n = 1 | n = 4 | n = 2 |
| K562-EGFR | EGFR only | 19.1 pM | - | 20.9 pM |
| | | n = 1 | | n = 1 |
| SKOV3 | Her2/EGFR | 1.4 pM | 11.7 pM | 13 pM |
| | | n = 1 | n = 1 | n = 1 |

Data are presented as mean +/- standard error (if n $\geq$ 2). Top: Binding of single antigen and dual antigen bridging proteins to single and dual-antigen positive cell lines. Binding was detected with anti-CD19 antibody FMC63 staining and the EC$_{50}$ values were derived from dose response flow cytometric analyses. Bottom: Bridging protein-mediated CAR-CD19 cytotoxicity against Her2-positive cell lines. CAR-CD19 T cells were used at an E:T ratio of 10:1. IC$_{50}$ values were derived from a titration of the bridging protein concentration in the assays. A 0 indicates that binding or activity were not detectable above background levels; a '-' indicates that the experiment was not performed.

CD19 ECD—anti-EGFR (#460, Table 1). A bridging protein targeting EGFR was also constructed (#416, S1 Table). Expression constructs were used to transfect 293T cells and the bridging proteins were purified. The binding affinities of single-antigen and the dual-antigen bridging proteins for the target antigens Her2 and EGFR were established using flow cytometric binding assays and Her2-positive, EGFR-positive or dual Her2/EGFR positive cell lines (S4 Fig). The dual-antigen targeting bridging protein #460 bound to the dual Her2/EGFR-positive cell line SKOV3 with an EC$_{50}$ value of 2.5 nM which was similar to the affinity obtained on the single antigen cell lines K562-EGFR and BT474 and the affinity seen using the single antigen binding bridging proteins (Table 3, S5 Fig).

Next, the potency of the multi-antigen bridging protein in mediating CAR-CD19 T cell cytotoxicity against Her2-positive, EGFR-positive and dual-antigen-positive cell lines was determined and IC$_{50}$ values were calculated. Dual antigen bridging protein-mediated cytotoxicity was highly effective against single antigen-positive cells, whether Her2-positive or EGFR-positive, with IC$_{50}$ values ranging from 5 pM—19 pM (Table 3, S6 Fig). Notably, the dual bridging protein was even more potent against dual-antigen positive cell lines, with an IC$_{50}$ values of 1.4 pM for SKOV3 cells in the presence of anti-CD19 CAR T cells (Table 3, S6 Fig). The single antigen bridging proteins CD19 anti-Her2 and CD19-anti-EGFR was at least 10-fold less potent (Table 3, S6 Fig).

## CD19-bridging proteins do not bind the CD19-cis ligands CD21 and CD81

CD19 binds CD21 in cis and reportedly binds tetraspanins such as CD81 [21, 22]. We used a panel of cell lines expressing CD20 (Raji), CD21 (Raji), CD81 (Raji, SKOV3, K562), and Her2 (SKOV3), or none of these antigens (U937), to evaluate potential trans-binding of the CD19-anti-Her2 bridging protein. Binding to all four cell lines was evaluated by flow

cytometry using an anti-His-tag antibody and the anti-CD19 antibody FMC63. As expected, the CD19 anti-Her2 bridging protein bound SKOV3 cells via the anti-Her2 scFv. The bridging protein did not bind to Raji cells (Her2-negative, CD21-positive, CD81-positive) as detected by anti-His-tag antibody staining (S2 Table). In order to rule out sequestration of CD21 and CD81 by endogenous CD19 in Raji cells we used OCI-LY3 cells (a lymphoma cell line) that do not express CD19. The bridging protein did not bind to OCI-LY3 (S2 Table). Representative flow cytometry data are presented (S7 Fig).

Detection of bridging protein with the anti-CD19 antibody FMC63 is possible for those cell lines which are CD19-negative: binding was observed with SKOV3 cells (Her2-positive, CD81-positive) but not with K562 cells (Her2-negative, CD81-positive) or U937 cells (negative for CD21, CD81 and Her2) (S2 Table, S7 Fig ). We previously showed that CAR-CD19 T cells do not kill SKOV3 cells via CD81 binding in the presence of the CD19-ECD control protein, but only in the presence of the CD19-anti-Her2 bridging protein [18]. These results show that the CD19 ECD cannot interact with either CD21 or CD81.

## Bridging protein secretion by CAR-CD19 T cells

To generate CAR-CD19 cells that secrete bridging proteins we cloned CD19-anti-Her2 sequences into a lentiviral vector downstream of a CAR-CD19 sequence and a P2A site to create a Her2-bridging CAR-CD19 (#374, Table 1). Viral particles were produced with a transducing titer of 1.8 x 2.5 x 10$^9$ TU/ml. T cells isolated from human donors were transduced with the lentiviral particles and analyzed for expression of the CAR-CD19 domain by flow cytometry. Using 6 different donors, we determined that the mean expression of the CAR-CD19 domain on the transduced primary T cells was 43% (range 24% - 63%, SD = 13%). Bridging protein secretion was measured by ELISA (FMC63 capture and anti-His detection). The mean secretion of the bridging protein was 40.2 ng/ml (n = 7 donors, duplicate points, range 10–81 ng/ml) in activated CAR T cell culture supernatants.

## Her2-bridging CAR-CD19 T cells have potent activity *in vitro*

Her2-bridging CAR-CD19 T cells were generated using normal human T cells (donor 45). The lentiviral transduction produced 47% CAR-positive T cells and the T cell pool was 38.5% CD8-positive and 99.5% CD3-positive (Fig 3A). The secretion of bridging protein was measured at 24 ng/ml in the activation assay (Fig 3B). Her2-bridging CAR-CD19 T cells were tested in cytotoxicity assays using either CD19-positive Nalm6 cells or Her2-positive SKOV3 cells at different E:T ratios; UTD T cells were used as a negative control. E:T ratios were calculated using the total T cell number and were not adjusted to reflect %-CAR positive cells. CD19-positive Nalm6 cells were potently killed by Her2-bridging CAR-CD19 T cells, showing that the bridging protein did not interfere with CD19 recognition by the CAR-CD19 domain (Fig 3B). Her2-positive SKOV3 cells were also potently killed by Her2-bridging CAR-CD19 T cells, demonstrating that the secretion of the bridging protein was sufficient to retarget the CAR-CD19 domain to SKOV3 cells. Potent cytotoxicity was seen down to an E:T ratio of 1:1 (Fig 3C).

These results show that Her2-bridging CAR-CD19 T cells can efficiently kill Her2-positive target cells. This is the first demonstration of cytotoxic elimination of CD19-negative tumor cells by retargeted CAR-CD19 T cells.

## Restimulation assay

Successful repetitive CAR T cell stimulation *in vitro* predicts potent CAR T cell activity *in vivo* [23], as repetitive stimulation models the waves of functional activity and cell proliferation

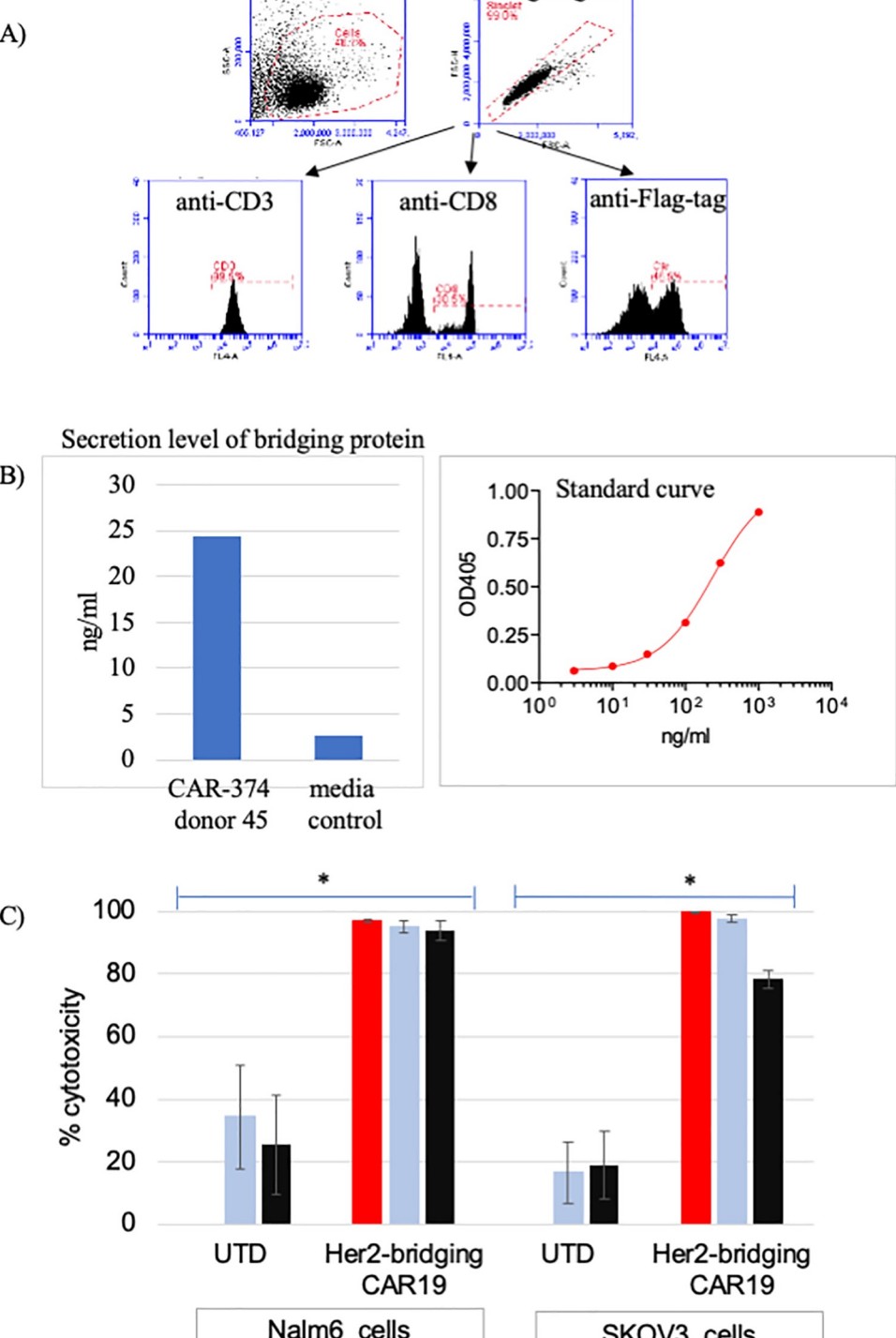

**Fig 3. Her2-bridging CAR-CD19 T cells are cytotoxic against both CD19-positive and Her2-positive cells *in vitro*.** (A, B) Example of the flow cytometry gating strategy (A) and the ELISA assay used (B) to determine the purity and phenotype of transduced human donor CAR-CD19 T cells that secrete a bridging protein. C) Her2-bridging CAR-CD19 T cells were added at different E:T ratios to Nalm6 cells (left) or SKOV3 cells (right) and cytotoxicity was measured. E:T ratios used were 10:1 (red bars), 3:1 (light blue bars) and 1:1 (black bars). The donor matched UTD cells are also shown at each E:T ratio.

needed for serial cytotoxicity. We investigated the impact of serial restimulation on Her2-bridging CAR-CD19 T cells using either Raji cells (CD19-positive lymphoma) or SKOV3 cells (Her2-positive ovarian carcinoma) or both, to model *in vitro* the potential encounters that the CAR T cell population would undergo in the patient setting *in vivo*.

One variable that complicates analysis of the re-stimulation assay results is the differential proliferative capacity of target cells during the course of the *in vitro* cell culture based cytotoxicity assay. Suspension-cultured B cell lines undergo rapid proliferation whereas attachment-cultured SKOV3 cells divide much more slowly and become contact-inhibited once confluent. In order to control for potential differential target cell proliferation in the assay we used mitomycin-c treatment to stop cell proliferation. Mitomycin-c is a double-stranded DNA alkylating agent that covalently crosslinks DNA, inhibiting DNA synthesis and cell proliferation. The cells treated were CD19-positive Raji lymphoma cells and Her2-positive SKOV3 ovarian carcinoma cells and the doses used were sublethal for at least 48 hours [24, 25].

Target tumor cells were treated with mitomycin-c, washed extensively, then cultured with CAR T cells for 48 hours. The CAR T cells were then washed and rested in media for 48 hours before being restimulated with a new preparation of mitomycin-c treated target cells. The process of stimulation, rest and restimulation was repeated through 4 cycles, through day 15. Her2-bridging CAR-CD19 T cells were stimulated repeatedly with either Raji cells or with SKOV3 cells. Her2-bridging CAR-CD19 T cells proliferated robustly in response to Raji cell restimulation through 4 rounds, as did CAR-CD19 T cells (Fig 4A). In contrast, Her2-bridging CAR-CD19 T cells did not increase dramatically above day 0 cell numbers when restimulated with SKOV3 cells, nor did CAR-Her2 T cells (Fig 4A). The differences in increase of CAR-CD19 T cells or Her2-bridging CAR-CD19 T cells in response to Raji cell restimulation versus no stimulation or SKOV3 restimulation were highly significant (p < 0.001, t-test for two independent samples for each comparison, n = 3). The cells restimulated with SKOV3 cells were viable, and they retained some cytotoxic activity through 3 rounds of stimulation (see below).

The Her2-bridging CAR-CD19 T cells recovered from each round retained cytotoxic activity (Fig 3B). In each cytotoxicity experiment the same number of CD3-positive CAR T cells were presented to luciferase-labeled target cells at a 5:1 E:T ratio. Representative results show that continual restimulation by Raji cells allows the Her2-bridging CAR-CD19 T cell population to maintain very high cytotoxic activity against both JeKo-1 cells (a B lymphoma cell line) and SKOV3 cells (Fig 4B). After 1 stimulation and after 3 stimulations, cytotoxic activity was sufficient to kill >95% of the target cells. In contrast, continual restimulation by SKOV3 cells gradually impacted the ability of the CAR T cells to mediate full cytotoxic activity. After 1 stimulation, Her2-bridging CAR-CD19 T cells were able to kill 100% of target tumor cells, whether JeKo-1 B cells or SKOV3 ovarian carcinoma cells (Fig 4B). After 3 rounds of SKOV3 cell stimulation, Her2-bridging CAR-CD19 T cells lost approximately 80% of cytotoxic activity (Fig 4B). Similar results were seen with the control CAR-CD19 T cells and the control CAR-Her2 T cells, ie. diminished cytotoxic capacity was only seen when T cells were restimulated by SKOV3 cells (Fig 4B). The difference in cytotoxic activity of CAR-374 cells when stimulated with Raji cells versus SKOV3 cells was significant at Round 3 whether the target cell was JeKo-1 or SKOV3 (p < 0.001, t-test for two independent samples for each comparison, n = 3).

## Bridging protein-secreting CAR-CD19 T cells are cytotoxic *in vivo*

Having established that Her2-bridging CAR19 T cells have cytotoxic activity against CD19-positive Nalm6 cells and Her2-positive SKOV3 cells *in vitro*, we tested the activity of these cells *in vivo*. Studies have been performed with anti-CD19 CAR T cells that secrete slightly different

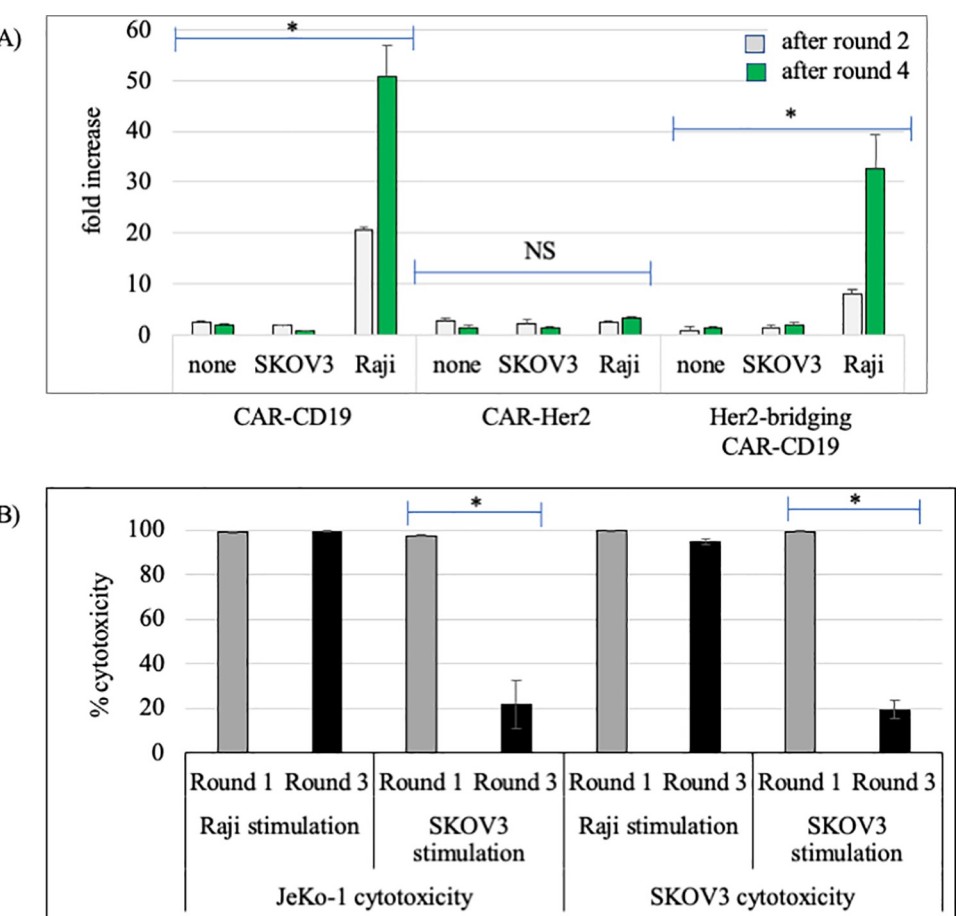

**Fig 4. CAR T cell activity in response to restimulation.** A) Extent of CAR T cell proliferation after 2 rounds and 4 rounds of Raji cell stimulation, SKOV3 cell restimulation or no stimulation; CAR T cell types are indicated below the x-axis. B) Extent of CAR T cell-mediated cytotoxicity against target tumor cells and 1 round and 3 rounds of Raji cell stimulation, SKOV3 cell restimulation or no stimulation. The round of stimulation, cell source of stimulation is indicated, and target cell in the cytotoxicity assay are indicated. * = p, 0.001. NS = no significant differences.

forms of the bridging protein (#142, S1 Table and #374, Table 1), with little apparent difference in outcomes. First, we analyzed the circulating half-life of purified bridging protein (#42) by injecting immunodeficient mice IV or i.p. with 200 μg of protein and analyzing protein concentration in the serum. The $T_{1/2}$ was determined to be ~1 hour after IV or i.p. adminstration (63 minutes and 65 minutes, respectively, S8 Fig). The short half-life was expected given the small size of the bridging protein and the absence of any half-life extension technology such as an Fc-domain, a PEG-domain or an anti-albumin binding domain. The half-life of the modified form of the bridging protein (#340) is not expected to be significantly different because no half-life extension technology (Fc domain or anti-albumin binder) was added.

We tested the activity of the CAR-CD19 and Her2-bridging CAR-CD19 T cells using xenografts in NOD-SCID-gamma common chain-deficient (NSG) mice. NSG mice injected iv with the CD19-positive cell line Nalm6 develop disseminated leukemia which is lethal approximately three weeks after injection. CAR-CD19 T cells effectively eliminated the *in vivo* growth and dissemination of Nalm6 cells as expected (S9 Fig).

To evaluate Her2-bridging CAR-CD19 T cells we injected $1 \times 10^6$ Nalm6-luciferase cells on day 1 and allowed the leukemic cells to expand for 4 days. The baseline degree of luminescence was established, mice were placed randomly into cohorts, and 3 different doses of CAR-T cells were injected iv. Mice were imaged weekly to determine the extent of disease burden. A dose response was observed for Her2-bridging CAR-CD19 T cells T cells with full eradication observed at the higher doses (Fig 5A). The treatment groups are also shown separately from the control groups in order to enhance visualization of the dose response (Fig 5B). This study demonstrates that Her2-bridging CAR-CD19 T cells recognize and eliminate CD19-positive cells *in vivo*.

Next we extended the *in vivo* modeling to the Her2-positive tumor cell line SKOV3. SKOV3 cells ($1 \times 10^6$) were implanted subcutaneously and allowed to establish. Mice were then randomized, the baseline degree of luminescence was established, and $1 \times 10^7$ CAR T cells were injected iv on day 14, at which time the tumors averaged 150mm$^3$ by caliper measurement. CAR-CD19 T cells (CAR-254) and Her2-bridging CAR-CD19 T cells were evaluated in this model. Mice were imaged weekly and caliper measurements of the tumor were taken twice weekly. By day 14 post CAR T cell injection the cohorts that received the Her2-bridging CAR-CD19 T cells had cleared the tumor mass as measured by caliper (Fig 5C). Tumor growth was significantly less than that measured in the negative-control cohorts (CAR-CD19 injected mice, or untreated mice (NA), Fig 5C). In a second study the *in vivo* activity of Her2-bridging CAR-CD19 T cells and CAR-Her2 T cells (CAR-390, Table 1) was compared and we found that the tumor control curves and the survival curves were indistinguishable (Fig 5D and 5E). All mice treated with Her2-bridging CAR19 T cells and CAR-Her2 T cells survived to day 42, while all animals in the negative control groups had to be sacrificed due to tumor burden.

In a followup study, serial rechallenge was evaluated. Her2-bridging CAR19 T cells were injected as described above and tumor growth was evaluated through day 42 (Fig 6A). All animals in the negative control groups (n = 30) had to be euthanized due to tumor burden; in contrast only one animal was lost in the Her2-bridging CAR19 T cell treated cohort (Fig 6B). The surviving animals in the Her2-bridging CAR19 T cell treated cohort were rested until day 55 (post tumor challenge) and then 4 mice were rechallenged with $1 \times 10^6$ Nalm6 cells to investigate whether the remaining CAR T cell pool could react to directly presented CD19 antigen. Three naïve NSG mice were also injected as a positive control for tumor engraftment and growth. The mice that had initially cleared the SKOV3 tumor effectively eliminated the Nalm6 cells, and there was no Nalm6 engraftment or expansion (Fig 6C). In contrast the naïve mice challenged with Nalm6 cells rapidly developed disseminated leukemia (Fig 6C) and were sacrificed by day 21 due to tumor burden.

These studies demonstrated that both mechanisms inherent to Her2-bridging CAR-CD19 T cells were active *in vivo*: the T cells could eliminate CD19-positive Nalm6 cells directly via the CAR-CD19 domain and also indirectly, via recognition of the CD19 ECD on the bridging protein that bound to the Her2-positive tumor cells.

## Discussion

Here we describe a novel approach to CAR-T cell therapy with three unique attributes: first, we demonstrate that it is possible to retarget CAR-CD19 T cells to kill any solid or hematologic tumor cell, exemplified here using Her2-positive solid tumor cells; second, we encode the CAR-CD19 domain and the CD19-bridging protein together within a lentiviral vector, such that it is secreted in parallel with expression of the CAR on the transduced T cell surface, thereby leveraging the inherent advantages of CAR-CD19 T cells; and third, we use modular

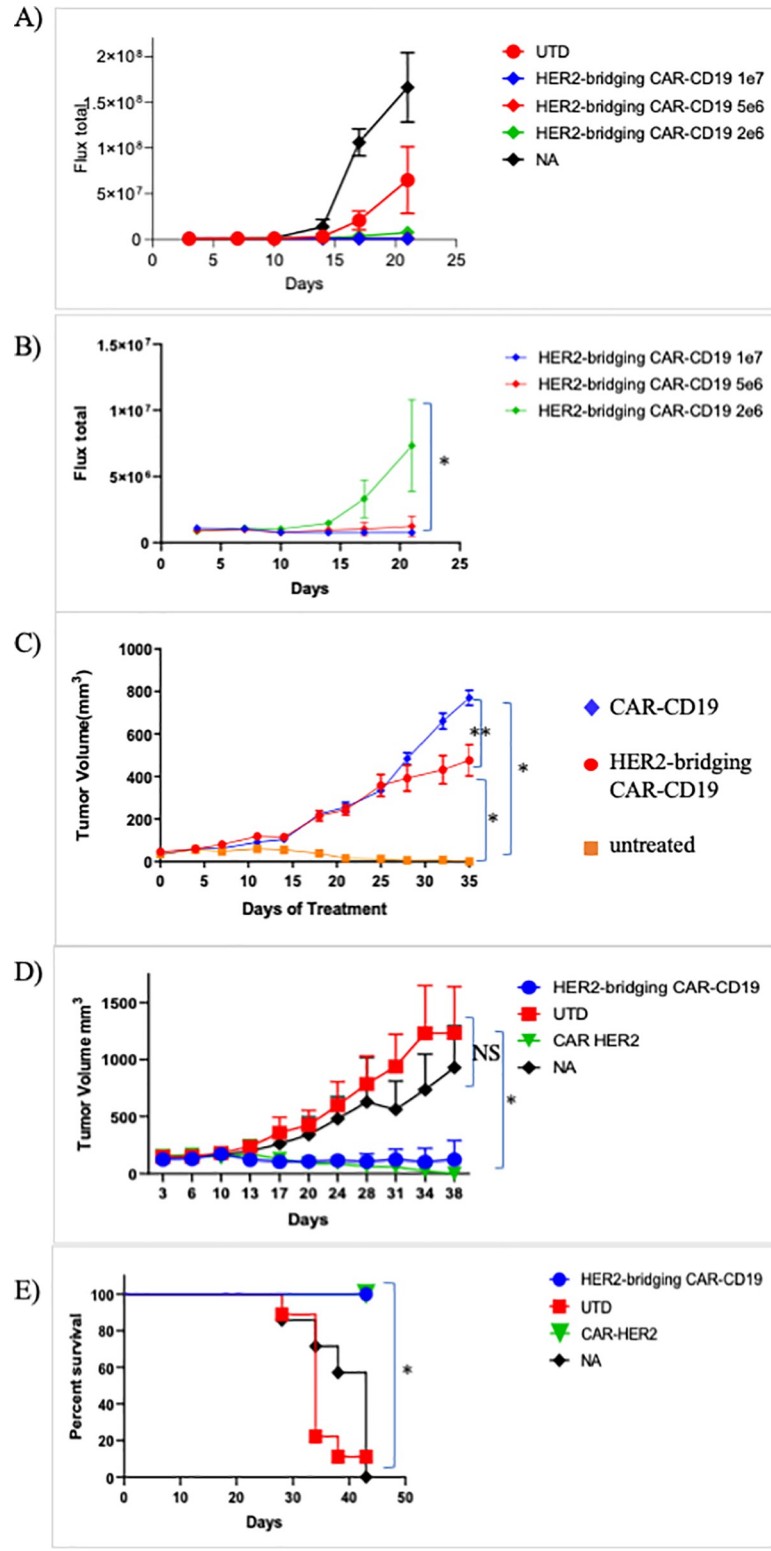

**Fig 5. Her2-bridging CAR-CD19 T cells are efficacious *in vivo*.** A) Dose responsive elimination of pre-established leukemia in an NSG mouse model was monitored after Her2-bridging CAR-CD19 T cells (CAR-142 and CAR-374, donor 36, 99.8% CD3-positive, 67% CD8-positivem, 59% Flag-tag-positive) were administered at doses ranging from 2 x 10⁶ to 1 x 10⁷ cells/mouse. Luminescence values for all cohorts (A) and the treatment cohorts (B) are shown. C-E) CAR T cells (donor 45, 99.2% CD3-positive, 41% CD8-positive, 38% Flag-tag-positive) were evaluated in a SKOV3

ovarian carcinoma model in NSG mice. C) CAR-142 T cells controlled SKOV3 tumor growth while control CAR T cells (CAR-254) had no effect. D) CAR-142 T cell control of SKOV3 tumor growth as compared to CAR-Her2 (CAR-390). CAR-142 and CAR-390 fully controlled tumor growth, while control mice succumbed and had to be euthanized by day 42 (panel E). * = p < 0.001. ** = p < 0.05. NS = no significant differences.

CD19-bridging proteins, which apply well understood protein-engineering principles and solutions to multiple limitations facing the CAR-T cell field, including providing a platform for multi-antigen targeting.

Robust *in vivo* expansion followed by prolonged CAR-T cell persistence is critical for the efficacy of CAR-T cell therapy directed to the B cell tumor antigen CD19 for the treatment of hematological malignancies [26–28]. In contrast, most solid tumor-targeted CAR-T cells appear to have poor persistence properties, irrespective of the antigen targeted. For example,

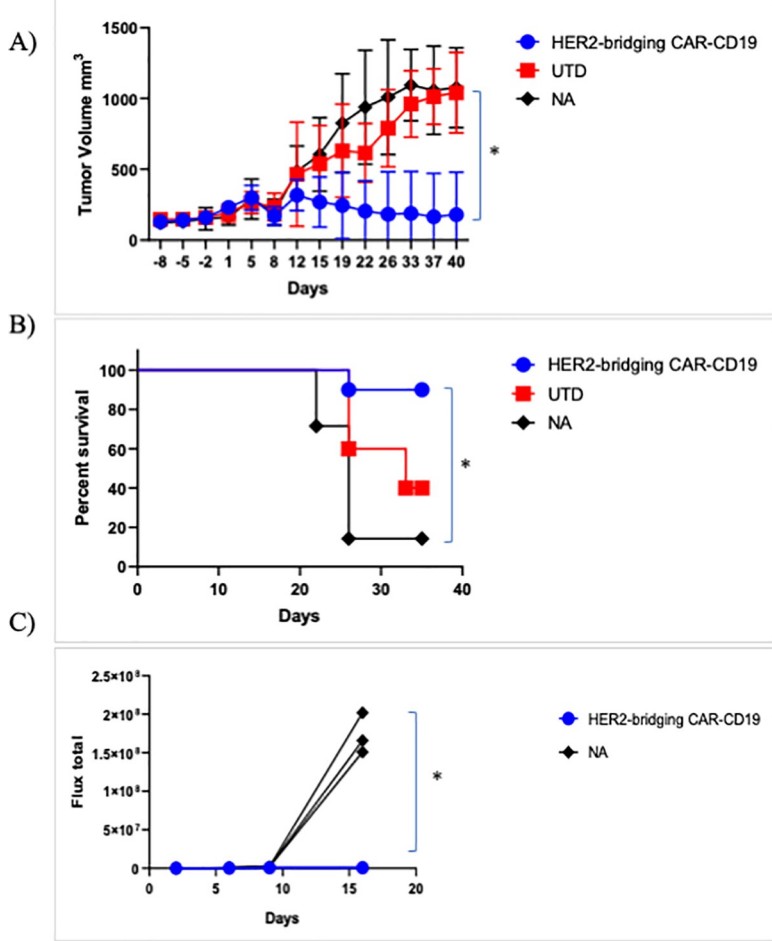

**Fig 6. Her2-bridging CAR-CD19 T cells eliminate Her2-positive tumors and CD19-positive tumors in a serial challenge *in vivo* model.** A-C) CAR T cells (donor 45, 99.2% CD3-positive, 45% CD8-positive, 34% Flag-tag-positive) were evaluated in a SKOV3 ovarian carcinoma model in NSG mice. A) CAR-374 treatment eliminated preestablished SKOV3 tumors in the NSG mouse model. B) 90% of CAR-374 treated mice survived past day 36, by which point the control mice were beginning to require euthanasia. C) CAR-374 that survived the SKOV3 challenge subsequently cleared CD19-positive Nalm6 leukemia challenge; all individual mice are shown. * = p < 0.001.

in a phase 1 clinical study of CAR T cell therapy for biliary tract and pancreatic cancers, Her2-targeting CAR-T cells showed peak expansion of 7-fold or less above baseline at day 9 post-injection, dropping thereafter [29]. Similarly, CAR-T cells targeting EGFRvIII that were administered for the treatment of glioblastoma persisted 21 days or less after infusion and had minimal clinical impact [30, 31]. A CAR-T cell therapy targeting IL13Rα2 reported sustained clinical impact in one patient despite minimal CAR cell persistence, and there were no other notable responses in that trial [32]. Notably, CAR T cells targeting PSMA expanded up to 500-fold over baseline following administration of $10^9$–$10^{10}$ total T cells to patients [33]. Expansion in patients appeared to be Il-2-dependent and lasted up to 4 weeks; several partial clinical responses were noted, but the duration of response was not reported. In general, poor CAR T persistence in solid tumors is associated with minimal clinical efficacy. Several interesting efforts to incorporate CARs into T cells that recognize viral antigens as a means of enhancing persistence have been advanced including a preclinical study using EBV-specific cytotoxic T cells to express an anti-CD30 CAR domain, and a clinical study utilizing HBV-specific T cells to express an anti-Her2 CAR domain [34, 35]. These therapeutics continue to advance in the clinic and it will be of keen interest to evaluate the expansion and persistence characteristics of these virus-specific T cells in patients.

Advances in the field focused on improving CAR-T cell activity include providing cytokine or chemokine support, preventing immunosuppression, and targeting intrinsic pathways to improve CAR-T cell fitness and persistence [36–39]. Recent efforts to provide artificial and immunologically favorable antigen presentation to CAR T cells also aim to improve CAR T cell fitness and persistence. For example an exogenously added CAR-T ligand was developed that binds to albumin, traffics to lymph nodes and is displayed by antigen-presenting cells (APC) [40]. CAR-T cells that encounter this ligand in the lymph node are therefore stimulated not only by the displayed antigen but also by costimulatory receptors and cytokines produced by the APCs. In another example, *in vivo* administration of a nanoparticulate RNA vaccine was used to deliver a CAR antigen into lymphoid compartments in order to stimulate adoptively transferred CAR-T cells. Again, presentation of the target antigen on APCs promotes immunologically productive activation and expansion of the injected CAR-T cells [41]. Notably, engagement of relevant stimulatory, chemokine and adhesion signals appears to favor the development of T cell memory [42], a critical element in long term immune protection. Finally, the importance of immunologically relevant T cell activation has been demonstrated in the immune checkpoint field, with defined roles for organized secondary and tertiary lymphoid organs becoming apparent [43–46].

In our system the CAR-CD19 T cell interaction with normal B cells will support production of immunologically relevant stimulatory signals including adhesion interactions, chemokine and cytokine signals, and costimulatory signals [47–50]. This organic presentation of immunologically relevant antigen does not require administration of additional agents or exogenous antigen, since B cells are APCs and represent a self-renewing source of CD19 that are present in lymphoid organs and in circulation. Moreover, in the *in vitro* experimental setting we showed that interaction of the Her-2 bridging CAR-CD19 T cells with a B cell target cell (Raji cells) had a distinctly different outcome than interaction with a solid tumor cell target (SKOV3), suggesting that at least some T cell supportive signals are present even if the target B cell is malignant. In the *in vivo* model, serial presentation of antigens (Her2, CD19) showed that the Her2-bridging CAR-CD19 T cells could engage both cell types.

Even in the setting of B cell leukemias and lymphomas, expression of CD19 on normal B cells may be critical for prolonged persistence of CAR T cells. In fully murine (syngeneic) models of CAR-CD19 therapy of B cell malignancy, control of tumor cell outgrowth over time and in the context of leukemic rechallenge required the presence of normal B cells [51, 52].

This observation suggests a unique role for normal B cells in providing a non-tumor dependent, self-renewing antigen source to support CAR-CD19 persistence. This concept is clinically validated, since some CAR-CD19-treated patients who have relapsed with CD19-negative tumors have persistent CAR-CD19 T cells in their circulation and continued B cell aplasia, demonstrating that the CAR-CD19 T cells remain functional [53]. An analysis of CAR-CD19 treatment of B cell leukemias found that the primary driver of CAR-CD19 T cell expansion and durable functional persistence was a cumulative burden of >15% of CD19 expressing leukemic and normal B cells in the bone marrow prior to lymphodepletion, and furthermore, that neither CAR T cell dose nor leukemia burden alone could predict CAR-CD19 T cell expansion and duration of engraftment [14, 54]. These results strongly suggest a critical role for normal CD19-positive B cells in supporting CAR-CD19 T cell expansion and persistence. In the setting of CAR-CD19 T cells secreting bridging proteins, CD19 expressed on normal B cells will be a readily accessed antigen source, to allow immunologically relevant CAR T cell expansion and to enhance persistence regardless of the targeted tumor antigen and indication.

B cell aplasia may be an important complication in some patients, however in the majority of patients, including most adults, B cell aplasia is a manageable toxicity. Hematology oncologists and rheumatologists have worked with B cell depletion agents for many years and the protocols for managing B cell aplasia are well understood [55–57]. It is conceivable that targeting B cells in the manner described will induce clinical toxicity such as cytokine release syndrome; such toxicities related to CAR T cell activity have also become increasing manageable [58, 59]. Finally, encouraging preclinical demonstration of the utility of off-the shelf agents such as dasatinib may lead to control over CAR T cell activity in the clinical setting [12, 60, 61].

The phenomenon of tumor antigen loss in response to CAR-T cell therapy is observed across indications and antigens and represents another major issue confronting adoptive cell therapy. The findings to date include the loss of target antigens from the cell surface in hematologic malignancies including ALL, NHL and multiple myeloma, and rapid tumor cell population escape due to the heterogeneity of antigen expression in solid tumors [62–64]. It is clear that countering resistance due to antigen loss and tumor antigen heterogeneity will require multi-antigen targeting solutions in many different cancer indications. Our use of CD19-bridging proteins provides a simple and pragmatic solution to this problem since CD19-bridging proteins can be readily produced in single-antigen, dual-antigen and multi-antigen targeting formats. As shown herein, a bi-specific CD19-bridging protein containing an anti-Her2 and an anti-EGFR scFv killed tumor cell expressing one or the other antigen, while showing exceptional potency when both antigens were present. We have used VHH domains in many of our constructs, and the linkage of two, three or more VHHs in series is well-established [65]. Thus, rapid, flexible and modular extension to three or more antigens is readily achievable with the CD19-bridging protein format, which builds on decades of established antibody and antibody fragment-based solutions [65, 66].

The adaptor CAR T cell field, in which antibodies, Fabs or other biologic constructs are used to link universal CARs to tumor antigens is an adjacent technology solution. Examples include CARs that target Fc receptors, peptides, haptens and leucine zipper binding [67]. The singular issue with such technologies is their reliance on tumor antigen expression, thus if the target antigen is lost the adaptor won't bind and the CAR cannot recognize the tumor cell. It is possible to create multiple adaptors to different antigens and use these in series or simultaneously, although regulatory and clinical translation may be slow.

There are alternative elegant solutions to dual antigen targeting, including tandem CARs, dual CARs, switch-gated CARs [68–70]; these are created via manipulation of the encoded CAR domain sequence, which then must be successfully expressed on the T cell surface. As

such technologies are extended to solid tumor antigens the expression of antigen on normal tissues will become a concern (see below), and may necessitate tuning of CAR affinities. Extension to three or more antigens while still retaining tumor targeting selectivity will require precise engineering. For example a recent paper described the use of defined ankyrin repeat domains (DARPins) to target three antigens successfully [71]. The technology was limited however as the potency of the trispecific CAR was lower than the corresponding monospecific CARs against tumor cells expressing a single target, revealing a potential limitation of multi-specific designs when expressed on the CAR T cell surface. Further, the issue of normal tissue cytotoxicity (eg. on EGFR) was not addressed. Thus the elegance of multi-CAR technologies is balanced by the difficulty, to date, of successfully engineering and expressing optimal constructs. In contrast, secretion or cell surface expression of proteins concurrent with CAR expression in CAR-CD19 T cells is well-established. This includes cytokines, receptors, and antibody domains, among others [37, 72]. Further, BiTE protein secretion from activated T cells or CAR-T cells does not activate, anergize or exhaust the T cells despite the fact that these proteins bind back to T cells via their anti-CD3 component [73]. For example, an anti-CD3 x anti-EGFR BiTE was successfully secreted from an EGFRvIII-directed CAR-T cell showing that such secretion does not interfere with T cell function [8]. Further, both EGFR and EGFR-vIII were specifically targeted in this system, demonstrating the utility of multi-antigen targeting from a single CAR T cell. In our system the secreted bridging protein likewise does not bind back to the CAR T cell in a manner that interferes with T cell function, and similarly enables multi-antigen targeting. The result is potent retargeted killing of the CD19-negative target cells with retention of cytotoxicity against CD19-positive cells.

Indeed, BiTEs and other forms of T cell engagers represent another adjacent technology solution, with many formats targeting diverse antigens under development. The major differentiation between the BiTE-like technologies and the CAR T engagers as described herein is specificity for the T cell. T cell engagers can bind to various anti-CD3-positive T cell subsets, to uncertain effect. Furthermore, several reviews have stressed the impotance of antigen-experienced T cell subsets in mediating T cell engager activity, posing a challenge in settings where such antigen-experienced cells are rare [73, 74]. In contrast the bridging proteins, as CAR T cell engagers, only bind to and activate the engineered cytotoxic CAR T cell, which we believe will be in sufficient concentration due to expansion from normal B cell engagement as well as antigen engagement via the bridging protein. Regardless, advances in the T cell and NK cell engager fields are interesting and will likely yield broader clinical application [75].

Our approach offers additional advantages. As a discrete entity, the secreted bridging protein has no obvious liabilities. The half-life is very short, which will limit systemic exposure and binding to normal tissues via the antigen-binding domains. Further, there is no Fc-domain that might mediate immune activation. The CD19 ECD is also essentially inert, having low potential immunogenicity [18] and no detectable binding to target cells expressing CD21 and/or CD81 that normally function, in cis, to mediate CD19 signaling within the B cell receptor complex. Finally, no detectable CAR-CD19-dependent killing can be seen in the presence of CD19 ECD alone [18]. We conclude that as a discrete entity, a bridging protein is functionally inert and has limited systemic exposure. Thus secreted bridging proteins should be biased to accumulate only where cell and antigen density is most abundant, ie. in the tumor itself, and should therefore primarily mediate activation of a CAR-CD19 T cell when bridging to an antigen-positive tumor cell. Such a system should provide a superior therapeutic index when compared to antigen-targeting CAR T cells.

One compelling translational strategy we are pursuing is to target Her2-positive CNS metastases in breast cancer, a leading cause of therapy failure. Systemically injected Her2-bridging CAR-CD19 T cells will expand in the presence of normal B cells, even if a patient remains

on anti-Her2 antibody therapy (eg. trastuzumab plus pertuzumab). Our ability to allow the patient to remain on standard of care therapy has two important consequences: 1) they are less likely to experience systemic tumor progression due to treatment interruption, and 2) the presence of trastuzumab will block systemically expressed Her2, making it less likely for the bridging protein to bind to any normal tissues expressing Her2. Of note, the antibody therapeutics used in Her2-positive cancer therapy (trastuzumab plus pertuzumab, or ADCs) cannot cross the blood-brain barrier effectively, and will not bind to Her2 in the CNS. In contrast, activated CAR-CD19 T cells readily access the CNS where they can find and control cancers [76, 77]. In the CNS the only Her2 present will be on the metastases, as normal brain and neuronal tissues are Her2-negative. Finally we note a recent paper showing that local activity of CAR-Her2 T cells in the presence of abundant tumor antigen did not cause systemic toxicity. In this singular case report, locally abundant rhabdomyosarcoma cells in the bone marrow supported CAR T expansion (following multiple CAR infusions) and 6 months of tumor control. Importantly, on-target/off-tumor toxicity was not reported [78]. This suggests that local accumulation of CAR T cells at the site of the tumor (here, in the bone marrow) improves the therapeutic index by limiting systemic exposure. These results are encouraging in the context of treating Her2--positive CNS metastases, as described above.

In summary, we present a universal method to retarget CAR-CD19 T cells to hematologic or solid tumor antigens, using CD19-bridging proteins secreted by the CAR-CD19 itself. This approach retains all the advantages of CAR-CD19 T cells, including optimal fitness and persistence and well a developed manufacturing history. In addition, the focus on protein engineering to create flexible and modular CD19-bridging proteins provides simple solutions to potency, antigen selectivity, and multi-antigen targeting.

## Materials and methods

### Human cell lines

SKOV3-luciferase (Cell Biolabs, San Diego, CA) were grown in RPMI containing 10% FBS and 0.8 mg/ml neomycin. BT474, Nalm6, K562, U937, Jeko-1, and 293T cells (ATCC, Manassas, VA) Raji cells (MilliporeSigma, St. Louis, MO), and 293FT cells (Thermo Fisher, Waltham, MA) were cultured according to supplier specifications. OCI-LY3 were cultured according to the suppliers instructions (DSMZ, Braunschweig, Germany). Stable cell lines expressing firefly luciferase were generated using lentiviral particle transduction and puromycin selection (GeneCopoeia, Rockville, MD).

**Protein expression constructs.** Genes were chemically synthesized by GenScript (Piscataway, NJ) and assembled into pcDNA3.1+ vectors (Invitrogen, Carlsbad, CA). The native CD19 ECD constructs were made with amino acids 1 to 278 (UniProtKB accession number P15391); eg. control construct #28. Construct #117 was made using the CD22 ECD amino acids 20–332 (UniProtKB accession number P20273 with a N101A mutation) and the anti-Her2 scFv sequence derived from trastuzumab [79]. Constructs #311 and #340 were made with stabilized CD19 ECD and the humanized anti-Her2 scFv (human 4D5 scFv, VH-G4Sx3-VL) [18]. These constructs are expressed at higher concentration than a previously decribed forms that contained the wildtype CD19 ECD (eg. as in #42) [18]. Construct #416 was made using a stabilized CD19 ECD and an anti-EGFR scFv sequence derived from panitumumab VH and VL sequences (Genbank accession numbers LY588610 and LY588596, respectively). Construct #460 contains the trastuzumab scFv the C terminal stabilized CD19 ECD-A, a second G4S linker, the panitumumab anti-EGFR scFv. It was produced by ligation of the pantitumumab fragment into a vector containing the trastuzumab-CD19 fusion. Constructs contained G4S linkers between functional domains and C-terminal 6xHis tags.

**K562-EGFR generation.** K562 cells expressing luciferase were transfected with a plasmid containing full-length EGFR (#OHu25437D, GenScript) using lipofectamine 2000 according to the manufacturer's protocol (Invitrogen). The cells were selected in G418 and clones isolated using limiting dilution.

## Expression of the bridging proteins

293T cells were transfected with expression constructs using lipofectamine 2000 following the manufacturer's protocol (Invitrogen, Waltham, MA). On day 3 post transfection the cell culture supernatants were harvested via centrifugation at 12,000 RPM for 4 minutes at 4˚C. His tagged bridging proteins were purified from the 293T transfected cell supernatant using a Ni Sepharose HisTrap excel column following the manufacturer's protocol (GE Healthcare, Marlborough, MA). The relevant fractions were pooled and buffer exchanged to PBS.

## CAR constructs

Construct #254 contains the FMC63 anti-CD19 CAR sequence (CAR-CD19). The anti-CD19 scFv sequence was derived from the FMC63 antibody [80], cloned in the VL-VH orientation, and includes a FLAG-tag, CD28 linker, transmembrane domain and cytoplasmic signaling domain (amino acids 114–220, UniProtKB accession number P10747) plus 4-1BB (amino acids 214–255, UniProtKB accession number Q07011) and CD3ζ (amino acids 52–164, UniProtKB accession number P20963) cytoplasmic signaling domains. The sequence was chemically synthesized (GenScript) and cloned into a derivative of the lentiviral vector, pCDH-CMV-MCS-EF1-Neo (Systems Biosciences, Palo Alto, CA) that contains an MSCV promoter. The CAR-CD19 sequence was also synthesized by Lentigen and cloned into their MSCV-contianing vector. Construct #390 (CAR-Her2) contains an scFv (Vh-Vl) from the anti-Her2 antibody FRP5 (GenBank accession number A22469) followed by the same components as the CAR-254 sequence (CAR-CD19: Flag tag, CD28 stalk transmembrane and cytoplasmic signaling domain and the 4-1BB and CD3ζ cytoplasmic signaling domains). The scFv portion of the CAR was chemically synthesized (Integrated DNA Technologies, Coralville, IA) and then assembled with a PCR fragment from the CAR-2 vector, minus the anti-CD19 scFv, using the NEBuilder Hi Fi DNA Assembly Master Mix (New England Biolabs, Beverly, MA). For the production of lentiviral particles, packaging plasmids pALD-VSVG-A, pALD-GagPol-A and pALD-Rev-A (Aldevron, Fargo, ND) and the expression plasmid were combined in Opti-MEM (Invitrogen). Trans-IT transfection reagent (Mirus, Madison, WI) was added. After incubation, the DNA/Trans-IT mixture was added to 293FT cells in Opti-MEM. After 24 hours, the media replaced with DMEM plus 10% FBS daily for 3 days and stored at 4˚C. The viral particles were precipitated by adding 5X PEG-IT (Systems Biosciences) incubating at 4˚C for 72 hours. The mixture was centrifuged at 3000 RCF for 30 minutes, the residual supernatant removed and the pellet resuspended in 200 μl PBS and stored at -80˚C.

## CAR-CD19 constructs containing bridging proteins

Expression cassettes for use in lentiviral constructs were chemically synthesized and cloned into a lentivirus vector containing an MSCV promoter (Lentigen Technology, Gaithersburg, MD). All constructs expressing bridging proteins contained the CAR-CD19 sequence described for CAR-254, a P2A cleavage site, and then the bridging protein sequences. Construct #374 contains the bridging protein sequence from construct #340. As such this bridging protein contains a stabilized CD19 ECD and replaced an earlier construct that used the wild-type CD19 ECD (#142); the activity of the two Her2-bridging CARs was very similar. Lentiviral particles were produced at Lentigen.

## CAR-T cell generation

For production and characterization of CAR T cells, CD3-positive human primary T cells were isolated from PBMC derived from normal human donors (Research Blood Components, Watertown, MA) and cultivated in ImmunoCult-XF T cell expansion medium (serum/xeno-free) supplemented with 50 IU/ml IL-2 at a density of $3 \times 10^6$ cells/mL, activated with CD3/CD28 T cell Activator reagent (STEMCELL Technologies, Vancouver, Canada) and transduced on day 1 with the CAR lentiviral particles in the presence of 1X Transdux (Systems Biosciences). Cells were propagated until harvest on day 10. Post-expansion, CAR T cells were stained with FLAG antibody to measure CAR expression. Briefly, $1 \times 10^5$ cells were incubated with anti-FLAG antibody (Thermo Fisher), diluted 1:100 in PBS for 60 minutes at 10°C, followed by a 1:100 dilution of anti-rabbit APC-conjugated antibody (Thermo Fisher). CAR T cells were stained for CD8 using anti-CD8 MEM-31 antibody, diluted 1:100 (Thermo Fisher). Cells were resuspended in PBS and fixed at a final concentration of 2% paraformaldehyde. Cell populations were analyzed using an Accuri C6 flow cytometer (BD Biosciences, Franklin Lakes, NJ).

## Analysis of bridging protein binding, specificity and affinity by ELISA

Plates were coated overnight at 4°C with 1.0 μg/mL anti-CD19 antibody FMC63 (#NBP2-52716, Novus Biologics, Centennial, CO) or 1.0 μg/mL purified Her2-Fc (#HE2-H5253, Acro-Biosytems, Newark, DE), in 0.1 M carbonate, pH 9.5. The plate was blocked with 0.3% non-fat milk in TBS for 1 hour at room temperature. After washing in TBST 3 times, the bridging protein was titrated from a starting concentration of 5 μg/mL using serial 3-fold dilutions in TBS/1% BSA and incubated 1 hour at room temperature. Detection with Her2-Fc, biotinylated Her2 (#He2-H822R, AcroBiosystems), or antibody FMC63 was performed by adding the detection reagent at 1.0 μg/mL and incubating for 1 hour at room temperature. The plates were then washed 3 times, followed by a 1 hour incubation with a 1:2000 dilution of anti-human IgG-HRP (#109-035-088, Jackson ImmunoResearch, West Grove, PA) for Her2-Fc detection, HRP-steptavidin (#21130, Thermo Fisher) for biotinylated Her2 detection, and anti-mouse IgG-HRP (#115-035-062, Jackson ImmunoResearch) for FMC63 detection. Then, 1-Step Ultra TMB-ELISA solution (#34028, Thermo Fisher) was added to develop the peroxidase signal, and the plate was read at 405 nm. Curves were fit using a four parameter logistic regression to calculate the $EC_{50}$.

## Flow cytometric analyses of bridging protein binding to cells and of cell surface antigen expression

When necessary, cells to be analyzed were detached with 0.5 mM EDTA in PBS. Tumor cells or lymphocytes were washed and analyzed in ice cold FACS buffer (PBS/1% BSA/0.1% sodium azide). Cells were resuspended ($5 \times 10^5/100$ μl) and purified bridging proteins (up to 10 μg/ml), or supernatants (100 μl) were added to 100 μl final volume and incubated with the cells at 4°C for 30 minutes. After washing, cells were resuspended ($5 \times 10^5/100$ μl) and incubated with detection antibody at 4°C for 30 minutes. The cells were resuspended in 100 μl total volume and stained with 5 μl anti-His-PE (#IC050P, R&D Systems, Minneapolis, MN) or anti-CD19 antibody FMC63-PE (#MAB1794, MilliporeSigma, Burlington, MA) for 30 minutes at 4°C, washed and resuspended using PBS, fixed with 1% paraformadehyde in PBS, and analyzed on the Accuri Flow Cytometer (BD Biosciences). Analysis of antigen expression by tumor cells was performed using flurochrome-conjugated monoclonal antibodies specific for the target antigen. Analysis of CAR T cell CD3 and CD8 expression was performed using PE-conjugated or APC-conjugated monoclonal antibodies specific for the target antigen.

## Analysis of secreted bridging protein concentration

To measure the secretion of bridging protein by CAR T cells, 600,000 cells were added at $3x10^6$ cells/ml to a 96-well U-bottom plate. Cells were activated with 1X ImmunoCult Human CD3/ CD28 T Cell Activator (#10971, Stemcell Technologies) and cultured for 4 days. The cells were briefly centrifuged at 300 x g for 4 minutes and the supernatants were collected. The amount of bridging protein was measured in an ELISA assay as described above.

## Cytotoxicity assays

Target tumor cell lines were stably transfected with a luciferase cDNA. Cells were seeded at $1x10^4$/25 µl cells per well in a 96 well round (suspension) or opaque flat (adherent) bottom plate in RPMI 1640 containing 10% FBS, without antibiotics. When included, dilutions of bridging proteins were made in 100 µl media and added to the cells. CAR T cells were thawed, washed once with media and recovered via centrifugation at 550 x g for 10 minutes at 4˚C. CAR T cells were added to the wells containing target cells to give an effector:target cell ratio as indicated, in a volume of 25 µl. In experiments using the standard CAR-CD19 T cells, bridging proteins were added as noted. The plates were incubated at 37˚C for 48 hours. For suspension cells, the plate was centrifuged at 550 x g for 5 minutes at room temperature, the pellet was rinsed with PBS, spun again, then lysed with 20 µl 1x lysis buffer (Promega). The lysate was transferred into a 96 well opaque tissue culture plate (Thermo Fisher). For adherent cell lines, the cell culture supernatant was removed, the cells were washed twice with cold PBS and then the 1x lysis buffer was added to the well. For the cytotoxicity assay the plates were read, post injection of luciferin from the detection kit, using a luminometer with an injector (Glomax Multi Detection System, Promega, Madison, WI). The percent killing was calculated based upon the average loss of luminescence of the experimental condition relative to the control wells containing only target cells.

## Order of addition assay

Target tumor cell lines Nalm6 and SKOV3 cells were seeded at $1 x 10^4$ cells per well in 50 µl in a 96 well round bottom plate for Nalm6 and 96 well white opaque plate for SKOV in RPMI 1640 containing 10% FBS without antibiotics. Proteins were applied at final concentration of 1 µg/ml in 25 µl media per well. CAR-CD19 T cells were thawed, washed once with media and recovered via centrifugation at 550 x g for 10 minutes at 4˚C. The CAR T cells were applied to the well at a 10:1 ratio to the target cells in 25 µl volume. Three conditions were applied as following: (1) the target cell, the bridging protein and the CAR T cells were added simultaneously, (2) the target protein and the bridging protein were preincubated at 37˚C for 10 minutes then the CAR T cells were added, (3) the bridging protein and CAR T cells were preincubated at 37˚C for 10 minutes then added to the target cells. The plates were incubated at 37˚C for 48 hours. The extent of cytotoxicity was determined as described above.

## Restimulation assay

On day 0, the CAR T and target cell cultures were set up at 2:1 (CD3+ CAR T: target cells at $2x10^4$: $1x10^4$ cells/well), and incubated for 48 hours with mitomycin C (MMC) treated Raji or SKOV3 cells, or without activating cells. The target cells were treated with 5 mg/ml MMC (#BP2531-2, Fisher Scientific) for 90 minutes at 37˚C and then washed twice with RPMI medium before adding to the T cells at a 2:1 ratio (CAR T:MMC treated cells). Before each restimulation, 100 µl of media was removed and 100 µl of MMC treated Raji or SKOV3 cells in fresh media were added on days 4, 8 and 12. Using this rest/restimulation sequence the CAR T

cells were re-stimulated every 4 days with mitomycin C (MMC) treated Raji or SKOV3 cells. Prior to each restimulation, parallel killing assays were set up on day 4 (after 1 round of restimulation) and day 12 (after 3 rounds of restimulation). For cytotoxicity assays the CAR T cells were counted, pooled, spun at 500 x g for 10 minutes and then resuspended at $5x10^5$ cells/ml. The cytotoxicity assay was set up as described above on JeKo-1 and SKOV3 cells at a 5:1 E:T ratio based on the total T cell count.

### *In vivo* modeling

Animal efficacy models were performed under an institutional IACUC protocol (Cummings School of Veterinary Medicine at Tufts University, Grafton, MA). Each individual study was approved by the Cummings School of Veterinary Medicine IACUC committee and assigned a specific protocol number to aid in tracking compliance in accordance with AAALC guidelines. For all studies the following criteria were used to monitor animal health and allow for humane euthanasia as needed. Lethargy, poor coat condition, poor body condition, abnormal ambulation, and weight loss (>10%) were general health parameters used to determine a euthanasia endpoint. In addition, in the subcutaneous solid tumor studies, mice were euthanized if the tumor mass reached 1500mm3. Any animals reaching any endpoint criterion were euthanized on the same day. No animals died prior to meeting the criteria for euthanasia.

The half-life of the wildtype CD19-anti-Her2 bridging protein was determined using SCID mice (n = 10) injected IV or IP with 10 mg/kg of protein. Blood samples were taken just prior to injection and at 30 minutes, 90 minutes, 6 hours, 24 hours, 48 hours, 72 hours, 96 hours and 120 hours. Whole blood was collected via the tail vein (100 μL per bleed). Whole blood was collected into EDTA K3 tubes (#411504105, Sarstedt, Numbrecht, Germany). The blood samples were spun for 10 minutes at 8500 rpm. The plasma was then transferred to a 1.5mL eppendorf tubes and frozen until analysis by ELISA as described above. ELISA calculations assumed a 2 ml mouse total blood volume of distribution. The half-life was calculated using the formula $N(t) = N0(1/2)t/t$ ½ where $N(t)$ is the quantity that remains, N0 is the initial amount, and t = time.

NOD/Scid x common-gamma chain-deficient (NSG) mice were purchased (Jackson Laboratories, Bar Harbor, ME) and allowed to acclimate in cage details for at least 3 days after receipt.

To establish a systemic leukemia model, NSG mice were injected IV with $1 \times 10^6$ Nalm6-luciferase cells on day one. Luciferase activity was monitored by the injection of luciferin (150mg/kg) followed by immediate whole animal imaging. Leukemia cells were allowed to engraft and expand for 4 days. The baseline degree of luminescence was established by imaging (IVIS 200, Perkin Elmer, Waltham, MA), and mice were randomized into experimental cohorts. There were 10 mice per cohort (n = 10) unless otherwise indicated. Mice were injected with CAR T cells on day 4 and imaged weekly to determine the extent of disease burden. Animals were sacrificed when tumor burden or animal health indicated, per the IACUC protocol.

To establish a solid tumor model we implanted $1 \times 10^6$ SKOV3-luciferase cells subcutaneously in the flank and allowed the tumors to grow to a palpable size of ~150mm$^3$ on average. Mice were then randomized and CAR T cells were injected IV the following day. There were 10 mice per cohort (n = 10) unless otherwise indicated. Caliper measurements of tumor volume were taken 2x weekly. Animals were sacrificed when tumor burden or animal health indicated, per the IACUC protocol. In the rechallenge model, mice that had cleared the SKOV3 tumors were injected ewith Nalm6 cells as described above.

## Supporting information

**S1 Fig. ELISA assays of bridging protein binding.** A) The capture agent was Her2-Fc and the detection agent was anti-CD19 antibody FMC63. B) ELISA assay of bridging protein binding:

the capture agent was anti-CD19 antibody FMC63 and the detection agent was biotinylated Her2. In both assays the bridging proteins (#42, #340) bound significantly better than the control proteins (#28, #117); $p < 0.01$ at the bridging protein $EC_{50}$ value.
(PDF)

**S2 Fig. Flow cytometric analyses of bridging protein binding.** A) Anti-Flag-tag staining of CAR-CD19 transduced donor primary T cells (donor 69, results are representative of n = 4 transduction experiments). B) Binding of purified CD19-anti-Her2 bridging proteins to anti-CD19 CAR transduced primary T cells (donor 69, n = 2, * $p < 0.004$ for binding of #42 vs #340). C) Expression of antigen Her2 on the ovarian carcinoma cell line SKOV3 (results are representative of n = 6 flow cytometry assays). D) Binding of bridging proteins to SKOV3 cells (triplicate wells, the data are from 1 of 2 experiments performed).
(PDF)

**S3 Fig. Binding of bridging proteins to SKOV3 and BT474 cell lines.** A) Representative gating of SKOV3 cells. B) Binding with 1 μg/ml of control and bridging proteins to SKOV3 cells. C) dose response binding to SKOV3 cells. D) Representative gating of BT474 cells. E) Binding with 1 μg/ml of control and bridging proteins to SKOV3 cells. F) dose response binding to BT474 cells. Detection of bound protein was with anti-CD19-PE antibody except for the CD22-anti-Her2 samples that were detected with anti-CD22-PE antibody.
(PDF)

**S4 Fig. Flow cytometric analysis.** Analyses of Her2 and EGFR expression on cell lines. A) K562-EGFR cells. B) BT474 cells. C) SKOV3 cells.
(PDF)

**S5 Fig. Representative bridging protein flow cytometry data.** A) Flow cytometry data showing various bridging proteins binding at saturation (A, C, E) and in dose response curves (B, D, F) to K562-EGFR cells (A, B), BT474 cells (C, D) and SKOV3 cells (E, F), as detected with anti-CD19 antibody FMC63-PE.
(PDF)

**S6 Fig. Bridging protein mediated cytotoxicity of EGFR-positive and Her2-positive cell lines.** A) K562-EGFR. B) BT474. C) SKOV3. Anti-CD19 CAR T cells (donor 54, 47% Flag-tag positive) were added at an E:T ratio of 10:1. Bridging proteins were added in a dose titration into the cytotoxicity assay.
(PDF)

**S7 Fig. Phenotypic analysis of protein expression, and CD19-anti-Her2 bridging protein binding to cell lines.** A) SKOV3. B) K562. C) Raji. D) U937. E) OCI-LY3.
(PDF)

**S8 Fig. Half-life measurement data.** PK measurements of a CD19-anti-Her2 bridging protein after injection into Rag-/- common gamma-/- mice.
(PDF)

**S9 Fig. Anti-CD19 CAR T cells control CD19-positive Nalm6 leukemic cell growth *in vivo*.** Mice were implanted with Nalm6 cell IV, then treated 4 days later with anti-CD19 CAR T cells (donor 54, 99.5% CD3-positive, 52% Flag-tag positive for CAR expression, and 56% CD8 positive by flow cytometric analyses). A) Representative images, day 12. B) Luminescence measurement, day 12. C) Luminescence measurement, day 19: note only 2 control (UTD) mice were still alive at day 19. The red bar marks the background luminescent reading. UTD lumin readings were statistically higher than treated mice on both days 12 and 19 $p < 0.001$. This

experiment is representative of 3 independent experiments.
(PDF)

**S1 Table. List of additional constructs used, with a description of the encoded sequences in column 2 and the format in which the expressed sequence is utilized in column 3.** The long dashes indicate a linker sequence. P2A refers to a cleavage sequence. BP refers to a bridging protein.
(PDF)

**S2 Table. Summary of bridging protein binding to cell lines.** Antigen expression is indicated as negative (-), dim (+), bright (++), or very bright (+++) or not done (nd). Binding of bridging proteins to cell lines was analyzed by flow cytometry as described in the text. The results are summarized as positive (+) or negative (-) for detectable binding.
(PDF)

## Author Contributions

**Conceptualization:** Roy R. Lobb, Paul D. Rennert.

**Investigation:** Christine Ambrose, Lihe Su, Lan Wu, Fay J. Dufort, Thomas Sanford, Alyssa Birt, Benjamin J. Hackel, Andreas Hombach, Hinrich Abken.

**Supervision:** Christine Ambrose, Benjamin J. Hackel, Hinrich Abken.

**Writing – original draft:** Roy R. Lobb, Paul D. Rennert.

**Writing – review & editing:** Christine Ambrose, Roy R. Lobb, Paul D. Rennert.

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
