## [Decision Letter · Decision Letter 0]

4 May 2020

PONE-D-20-08788

CD19-targeting CAR T cells potently redirected to kill solid tumor cells

PLOS ONE

Dear Dr. Rennert,

Thank you for submitting your manuscript to PLOS ONE. After careful consideration, we feel that it has merit but does not fully meet PLOS ONE’s publication criteria as it currently stands. Therefore, we invite you to submit a revised version of the manuscript that addresses the points raised during the review process.

Dear colleagues, after reading the reports of the reviewers I have to ask you to improve your ms according to their comments. My major personal view is also summarized below. In its present form the ms is hard to follow and to understand. However, rewriting, reorganization, adding helpful schematic views explaining which construct was actually used and why you used it in the respective experiment, and why you switched etc. might be a way to simplify the reading of your ms. 

We would appreciate receiving your revised manuscript by Juli 2020. To enhance the reproducibility of your results, we recommend that if applicable you deposit your laboratory protocols in protocols.io, where a protocol can be assigned its own identifier (DOI) such that it can be cited independently in the future. For instructions see: http://journals.plos.org/plosone/s/submission-guidelines#loc-laboratory-protocols

We look forward to receiving your revised manuscript.

Kind regards,

Michael P. Bachmann, Ph.D.

Academic Editor

PLOS ONE

Journal Requirements:

"Animal research was conducted following the IACUC guidelines and protocols of the Cummings School of Veterinary Medicine at Tufts University. Anesthesia was

performed using isoflurane. Euthanasia was performed under observation, using CO2

and methods that ensured minimal distress."

Please amend your current ethics statement to confirm that your named ethics committee specifically approved this study.

For additional information about PLOS ONE submissions requirements for ethics oversight of animal work, please refer to http://journals.plos.org/plosone/s/submission-guidelines#loc-animal-research  

Once you have amended this statement in the Methods section of the manuscript, please add the same text to the “Ethics Statement” field of the submission form (via “Edit Submission”).

6. Thank you for stating the following in the Financial Disclosure section:

"The authors received no specific funding for this work."

We note that one or more of the authors are employed by a commercial company: "Aleta Biotherapeutics"

Additional Editor Comments (if provided):

Dear Authors,

your ms was reviewed by four independent reviewers resulting in more or less similar or at least related comments. One major problem that came up in almost every report is that the ms is somehow difficult to read and confusing. In my personal opinion this is mainly caused by the use of different constructs without giving sufficient rationale for switching between them which finally makes it quite difficult to understand and follow the overall concept of the ms. Somehow a survey of the constructs would be helpful to easily identify which construct was actually used, what are the differences and possible consequences, and for what reason or purpose you selected another one in the respective experiment. Perhaps adding schematic views and giving in more detail and arguing why which construct was actually used could already solve this problem. In summary, rewriting, reorganizing the ms thereby responding the comments of the reviewers might helpful to overcome their major issues.

I hope that the reviewer reports will be helpful for you to carefully modify your ms accordingly.

with my best regards

yours sincerely

Michael Bachmann

Reviewers' comments:

Reviewer's Responses to Questions

**Comments to the Author**

1. Is the manuscript technically sound, and do the data support the conclusions?

Reviewer #1: Partly

Reviewer #2: Partly

Reviewer #3: Partly

Reviewer #4: Partly

2. Has the statistical analysis been performed appropriately and rigorously? 

Reviewer #1: N/A

Reviewer #2: I Don't Know

Reviewer #3: No

Reviewer #4: N/A

3. Have the authors made all data underlying the findings in their manuscript fully available?

Reviewer #1: Yes

Reviewer #2: Yes

Reviewer #3: No

Reviewer #4: No

4. Is the manuscript presented in an intelligible fashion and written in standard English?

Reviewer #1: Yes

Reviewer #2: No

Reviewer #3: No

Reviewer #4: Yes

5. Review Comments to the Author

Reviewer #1: • Using already approved anti-CD19 CAR T cell therapies as foundation to set up CD19-ECD-anti-tumor-scFv to redirect CD19-CAR T cells to solid tumors is a smart approach since it also may enhance persistence of such T cells in the body. Yet, the concept has been published before by the same group and is also somewhat similar to the concept of Malcom Brenner´s group published in 2007 (Blood, 2007, 110:2620-2630), where latent EBV infection was suggested as persisting source for stimulating EBV-specific T cells modified with anti-CD30 CAR. The authors should mention this publication in their discussion. However, there are some novel aspects described in the submitted manuscript, in particular an improved re-stimulation of CD19-CAR-T cells simultaneously modified with secreted CD19 binding protein.

General remarks

• In times of evolving viral threats: Why employing a system to eradicate tumors, which knock out the B cell compartment and therefore potential virus-neutralizing antibody responses? The authors should include limitations of their approach (including obvious regulatory hurdles).

• The manuscript is confusing…to much data which doesn´t really help to understand the manuscript… please more stringent presentation (for example present data containing only the improved CD19-anti-HER2 binding protein (#340) instead of #42 and #340, skip data showing less suitability of BP aggregates.

In almost all datasets, the number of replicates or independent experiments is not clear to the reader.

Major points

The authors performed transductions of T cells resulting in different transduction efficiencies. They did not sort CAR-positive T cells nor did they perform antibiotic selection of transduced T cells. However, they used specific effector to target ratios in the cytotoxicity assays depicted in figures 1 and 2. Please clarify, if the effector to target ratio is calculated by the number of T cells or by the number of CAR-positive T cells, the latter resulting in a significant number of non-transduced T cells in the experiments. How were the non-transduced T cell (UTD control) treated, how was their number adjusted in the experiments? The authors should clarify the points.

What is the purity of the expanded CAR-T cell populations? Have the authors ruled out NK cell or B cell contaminations in all experiments? It is advised to provide data showing purity of the expanded/restimulated immune cells. Furthermore, since re-stimulation of CD19-CAR T cells equipped with ectopic binding protein expression or recombinant binding protein is the centerpiece of the manuscript and might have clinical impact, it is recommended to recapitulate expansion of T cells transduced with #374, #416, #460 using autologous B cells as source for CD19. It is furthermore recommended to demonstrate the selective expansion of CAR-positive T cells (increase in percentages in the T cell population) upon stimulation with CD19+ target cells and the absolute increase in CAR-T cell numbers (please provide absolute numbers and no normalized values as depicted in Fig. 3.).

Another concern relates to the cytotoxicity assay chosen for the experiments. The authors employs Luc-modified target cells for analysis. Target cells were confronted with effector cells and binding proteins for 48h. Such a setting should result in exaggerated Luc activity in control cells due to cell doublings when compared to treated target cells, which might have succumbed to cells death during the first hours of the assay. Likewise, this results in an overstated cytotoxicity of effector cells towards target cells. It is recommended to shorten the incubation time of the cytotoxicity assay to 18- 24 h. Calculate specific lysis of cells by including spontaneous(minimal) lysis and maximal lysis of untreated target cells and to calculate specific cytotoxicity to the formula specific cytotoxicity = 100 × (Luc activity target cells − Luc activity minimal release) / (Luc activity maximum release − Luc activity minimal release). In this regard, please provide data, showing that treatment with CD19 bridging proteins alone to HER2 and EGFR or supernatants derived from T cells transduced with # 142 and #374 do not affect proliferation or viability of antigen-positive target cells. Therefore, the authors are strongly encouraged to exclude a potential therapeutic effect of the bridging proteins, which also might have implications for in vivo experiments.

The authors suggest that the CD19-anti-HER2 bridging protein does not compromise CD19-CAR- function. Yet, when adding a competing ligand for a CAR most readers likely anticipate that the cytotoxicity of CD19-CAR T cells towards CD19+ NALM6 target cells should be attenuated. The authors should give reasons why a competing CD19 bridging protein does not inhibit CD19-CAR T cell cytotoxicity towards NALM6 cells (i.e. crosslinking on target cells increasing avidity etc.).

Minor points:

Fig.1D and E, what is the effector to target ratio used for the experiments?

Supplemental Figure S3 A, for clarification,please provide dot blots showing double staining of CAR and bound #42 and #340 recombinant proteins, respectively.

Reviewer #2: See attachment for Review Comments to the Author

...................................................................................................................................................................

Reviewer #3: Broad clinical translation of CAR T cell technology beyond CD19 malignancies is hampered by many factors e.g. “on-target, off-tumor” toxicities, antigen-loss or poor CAR T cell persistence, expansion. In the submitted manuscript, Ambrose et al. present a novel strategy how conventional CD19 CAR T cells can be repurposed for killing of CD19-negative cancer cells (exemplified on Her2+ solid tumors) and thereby might overcome some general limitations. For this purpose, T cells were engineered to simultaneously express a CD19 CAR and soluble bridging proteins (BP) composed of the CD19-ECD and a tumor-specific (anti-Her2) targeting moiety. These T cells were able to kill both CD19+Her2- B cell lines and Her2+CD19- solid tumor cell lines both in vitro and in vivo.

Even though the principle idea is interesting, I do not recommend publication of the manuscript in the current version mainly due to the following reasons:

The numbering of CAR constructs and bridging proteins hampers enormously the understanding of the presented data. It is not clear, why the molecules were not numbered in a clear ascending order (#1, #2, #3). For publication, nomenclature must be easier especially for understanding the Figures e.g. by giving short names (e.g. CD19ECDwt-antiHer2, CD19ECDA-antiHer2,…) rather than confusing numbers (even though they are explained in Table 1). In my opinion, the manuscript would benefit a lot from a schematic overview of all CAR constructs and BP. They should be also listed in separate Tables/graphs.

The authors did not mention nor discuss one major limitation of their approach: attack of B cells in tumor patients which will result in long-term B cell aplasia and might also bear the risk for CRS, CRES as seen in B cell lymphoma and leukemia patients.

Several questions remain concerning assay design and data interpretation:

o page 8: “….the CD19 stabilized bridging proteins #311 and #340 bound SKOV3 cells…” Why do the authors here include #311 and not in the tests before? #311 was also not applied in the following cytotoxicity tests to compare functionality with #42 and #340.

o Page 9: “…in additional flow cytometry assays (n= 2 to 6 repeats) using Her2-positive SKOV3…” Why was the experiment repeated? Binding results for the bridging proteins to SKOV3 were already presented in Figure S3D. What does “n= 2 to 6 repeats” mean? Please identify how often a certain assay was performed (might be included in Table 2)!

o Figure 1A+B: The authors should include appropriate mock transduced control T cells, to confirm specificity of their system. Furthermore, it is not clear which CAR T cell was used as control in Fig1B (diagonal bars). For cytotoxicity assays with SKOV3, authors should include controls with CD19 CAR T cells + tumor cells without the bridging protein. At which E:T ratios were the assays performed?

o To characterize functionality of BP-secreting CD19 CAR T cells also cytokine secretion and proliferation/T cell expansion (not only in the context of restimulation) should be analyzed.

o Supplemental Table S1: The authors cannot draw the conclusion that CD19 ECD doesn´t bind to CD21 as the selected cell line Raji naturally expresses CD19 that might already block the CD21 binding side.

o Restimulation assay: How did the authors removed tumor cells from co-cultures to rest CD19 CAR T cells before the next round of restimulation? For data interpretation, it would be of interest to analyze memory phenotype, T cell exhaustion and activation status of CAR T cells during the restimulation process.

o Figure 3C shows that �Her2 CAR T cells do not lose cytotoxic capacity against SKOV3, after restimulation with B cell lines, but after restimulation with SKOV3. This indicates that the cell lines used for restimulation may provide “general” stimulatory or inhibitory signals to all CAR T cells (independent of their specificity). To analyze this in more detail, not restimulated CAR T cells should be included into the cytotoxicity assays. In addition, all restimulation assays should be performed with other CD19+ and Her2+ cells. It would be even more relevant to show that HLA-matched human B cells from peripheral blood (and not an artificial cell lines) are able to restimulate CD19 CAR T cells as this reflects the situation in the patients.

o Cytotoxicity assay with the dual bridging protein anti-Her2-CD19ECD-anti-EGFR was performed in comparison to #311. Why did the authors not selected #42 as control. This would be more reasonable as #460 was constructed on the basis of #42. Selection of different CD19ECD variants might influence overall performance redirected CD19 CAR T cells. No data are available showing that #311 and #42 mediate comparable tumor lysis.

o Why was in vivo serum half-life only analyzed for #42, but not for #340? Why do authors switch between #142 and #374 during their experiments?

o In vivo functionality of BP-expressing CARs (#142, #374) should be compared to a group that received CD19 CAR T cells (#254) and BP (#42, #340) as infusion.

Data presentation:

• Statistical analysis is missing in most of the Figures.

• Please indicate number of performed experiments.

• Abbreviations used in the Figures should be explained in the figure caption e.g. NA, UTD

• Table 1: #311 description is incorrect

• Figure 1C/D: It is stated that a “fixed E:T ratio” was used, please indicate exactly at which E:T ratio the assay was performed 10:1, 3:1, or 1:1?

• Page 7: “When the two bridging protein #42 and #340, were tested…in an ELISA (FMC63 capture, Her2-Fc detection)…”  In Table 2, combination of FMC63 capture + Her2-Fc detection is indicated as “ND” by the authors. What was actually done? Are these data presented in Supplemental Figure S2? If so, authors should also refer to the Figure in the main text.

• Figure 5 consists of three panels (a-c). However, in the text the authors refer to Figure 5d that does not exist.

• Figure S3B: Authors state that binding affinity of #42 and #340 is significantly different.

Please indicate test and significance level in figures

What does MW(Kd) mean? Do the authors here refer to “Kilodalton = kDa”?

• Figure S3A/C: What is the purpose of adding A03, A02, A01 or D12, C12, B12 in the figure

legend?

Discussion:

• Why is the secretion of BP via CD19 CAR T cells beneficial compared to BP infusions? With regard to “on-target, off-tumor” effects, the system …may necessitate tuning of CAR affinities…”, as stated by the authors. Multi-specific tumor-targeting with CD19 CAR T cells secreting BP will also require production of several T cell products from the same patients.

• The proposed approach should be compared to other adaptor CAR systems described in the literature.

Reviewer #4: In the manuscript the authors describe an approach to expand the target antigen landscape for “classical” CD19 CAR-T by providing soluble bridging adaptors enabling the CD19 CAR to recognize other tumor antigens than CD19 and kill antigen-carrying target cells in vitro and in vivo.

In general, the manuscript is well written and straight forward. However, the experimental data presented is sparse and critical experiments are missing. Furthermore, the authors do not discuss the pro and cons of their approach in light of published BiTE and switch CAR-T approaches.

Result section

Autoactivation of CD19 CAR-T by the soluble binding proteins is a critical issue for the system. Data (e.g. Absence of cytokine secretion) should be provided to demonstrate the safety of the system as claimed in the introduction and discussion section.

Table 2. Indicate on how many repeated experiments and technical repeats the affinity determination is based on.

Stability at 37°C is only relevant if demonstrated in presence of human serum – was human serum present? Please comment.

Determined half-life of the bridging proteins is ~ twice as long as the half-life of clinical bispecific antibodies (e.g. Blinatumomab). Maybe the ELISA used is only sensitive for dimers? Was the impact of mouse serum on the ELISA evaluated? The authors should comment on this.

The used coding system for the CARs and bridging proteins makes it difficult to follow the manuscript. The reviewer suggests to replace it by using short names indicating antigen combination, e.g. #42 = CD19-HER2. In all figures the number of tested donors and mice are missing. The information is provided in the M&M section, but should be included in the figure legends. F

Please comment on the high background lysis of non-transduced cells in Fig. 1 and 2. In Fig. 2 the UTD control seems to be missing.

Figure 4 is miss-labelled as Figure 5

Statistical analysis of the in vivo data is missing

The re-challenge in vivo experiment is based on a very small number of mice, expanding the groups would strengthen the data set.

Discussion section

Targeting less differentiated antigens beyond CD19, CD22, or BCMA bears the risk of on target/of tumor toxicities. The described approach would rely on continuous secretion of the bridging protein. How does the authors envisioned to avoid on target/off tumor toxicities? Claimed “limited systemic exposure” is very hypothetical. In particular the reviewer would expect sever toxicities for a clinical approach of the HER2-specific adaptor due to a first pass effect in the lung as seen with CAR-T harboring the same binder (Morgan et al. 2010).

The approach hazard the consequences of a continued B cell aplasia, which is clinical manageable, however bears great health risks for patients, comes along with high costs and has a severe adverse impact on the quality of life of patients. The authors should comment how they envision implementing such an approach into the clinical practice.

In the light of these arguments the approach falls back beyond switch CAR-T approaches published by several groups, in which the CAR binding moiety does not recognize a surface protein. The authors should at least cite the most relevant publications from these field and compare their approach to these earlier established systems of which some are already clinical evaluated.

Minor comments:

p. 8 Donor human T cells were transduced with lentiviral particles that express the scFv derived from the anti-CD19 antibody FMC63 (CAR-CD19, encoded by construct #254, Supplemental Fig. 3a). -> Please clarify that the lentiviral vector encoded transgene is a CAR harboring the scFv derived from the anti-CD19 antibody FMC63

Some typing errors throughout, e.g.

p. 13: Bindng of CD19-anti-Her2 bridging proteins

p. 25: In the in vivo model, serial presentation of antigens (Her2, CD19) showed that the Her2-bridging CAR-CD19 T cells could ehgage both cell types.

6. PLOS authors have the option to publish the peer review history of their article (what does this mean?). If published, this will include your full peer review and any attached files.

Reviewer #1: No

Reviewer #2: No

Reviewer #3: No

Reviewer #4: No

---

## [Author Response · Author response to Decision Letter 0]

4 Aug 2020

The responses are in the cover letter and in the response to reviewers. I am not sure I need to add anything here?

---

## [Decision Letter · Decision Letter 1]

12 Feb 2021

Anti-CD19 CAR T cells potently redirected to kill solid tumor cells

PONE-D-20-08788R1

Dear Dr. Rennert,

We’re pleased to inform you that your manuscript has been judged scientifically suitable for publication and will be formally accepted for publication once it meets all outstanding technical requirements.

Kind regards,

Joe W. Ramos, Ph.D.

Academic Editor

PLOS ONE

Additional Editor Comments (optional):

Reviewers' comments:

Reviewer's Responses to Questions

**Comments to the Author**

1. If the authors have adequately addressed your comments raised in a previous round of review and you feel that this manuscript is now acceptable for publication, you may indicate that here to bypass the “Comments to the Author” section, enter your conflict of interest statement in the “Confidential to Editor” section, and submit your "Accept" recommendation.

Reviewer #1: All comments have been addressed

Reviewer #5: (No Response)

2. Is the manuscript technically sound, and do the data support the conclusions?

Reviewer #1: Yes

Reviewer #5: Yes

3. Has the statistical analysis been performed appropriately and rigorously? 

Reviewer #1: Yes

Reviewer #5: Yes

4. Have the authors made all data underlying the findings in their manuscript fully available?

Reviewer #1: Yes

Reviewer #5: Yes

5. Is the manuscript presented in an intelligible fashion and written in standard English?

Reviewer #1: Yes

Reviewer #5: Yes

6. Review Comments to the Author

Reviewer #1: Christine Ambrose and coauthors have addressed my concerns and critics in a satisfactory manner. I would rate the manuscript suitable for publication in PLOS ONE.

Reviewer #5: This is a very thorough demonstration of how secretion of bridging protein may help overcome the challenges we have had with CAR T therapy against solid tumors. They have also addressed concerns for T cell persistence, so I would be interested to see how this model of CAR T cell work in patients.

7. PLOS authors have the option to publish the peer review history of their article (what does this mean?). If published, this will include your full peer review and any attached files.

Reviewer #1: No

Reviewer #5: No

---

## [Editor Report · Acceptance letter]

10 Mar 2021

PONE-D-20-08788R1 

Anti-CD19 CAR T cells potently redirected to kill solid tumor cells 

Dear Dr. Rennert:

I'm pleased to inform you that your manuscript has been deemed suitable for publication in PLOS ONE. Congratulations! Your manuscript is now with our production department. 

Kind regards, 

on behalf of

Dr. Joe W. Ramos 

Academic Editor

PLOS ONE